# TRAINING-FREE DEFENSE AGAINST ADVERSARIAL ATTACKS IN DEEP LEARNING MRI RECONSTRUCTION

## ABSTRACT

Deep learning (DL) methods have become the state-of-the-art for reconstructing sub-sampled magnetic resonance imaging (MRI) data. However, studies have shown that these methods are susceptible to small adversarial input perturbations, or attacks, resulting in major distortions in the output images. Various strategies have been proposed to reduce the effects of these attacks, but they require re-training and may lower reconstruction quality for non-perturbed/clean inputs. In this work, we propose a novel approach for mitigating adversarial attacks on MRI reconstruction models without any retraining. Based on the idea of cyclic measurement consistency, we devise a novel mitigation objective that is minimized in a small ball around the attack input. Results show that our method substantially reduces the impact of adversarial perturbations across different datasets, attack types/strengths and PD-DL networks, and qualitatively and quantitatively outperforms conventional mitigation methods that involve retraining. We also introduce a practically relevant scenario for small adversarial perturbations that models impulse noise in raw data, which relates to *herringbone artifacts*, and show the applicability of our approach in this setting. Finally, we show our mitigation approach remains effective in two *realistic* extension scenarios: a blind setup, where the attack strength or algorithm is not known to the user; and an adaptive attack setup, where the attacker has full knowledge of the defense strategy.

## 1 INTRODUCTION

Magnetic resonance imaging (MRI) is an essential imaging modality in medical sciences, providing high-resolution images without ionizing radiation, and offering diverse soft-tissue contrast. However, its inherently long acquisition times may lead to patient discomfort and increased likelihood of motion artifacts, which degrade image quality. Accelerated MRI techniques obtain a reduced number of measurements below Nyquist rate and reconstruct the image by incorporating supplementary information. Parallel imaging, which is the most clinically used approach, leverages the inherent redundancies in the data from receiver coils (Pruessmann et al., 1999), while compressed sensing (CS) utilizes the compressibility of images through linear sparsifying transforms to achieve a regularized reconstruction (Lustig et al., 2007; Jung et al., 2009). Recently, deep learning (DL) methods have emerged as the state-of-the-art for accelerated MRI, offering superior reconstruction quality compared to traditional techniques (Hammernik et al., 2018; Knoll et al., 2020a; Akçakaya et al., 2022). In particular, physics-driven DL (PD-DL) reconstruction has become popular due to their improved generalizability and performance (Hammernik et al., 2018; Aggarwal et al., 2019).

While PD-DL methods significantly outperform traditional MRI reconstruction techniques, these approaches have been shown to be vulnerable to small adversarial perturbations (Goodfellow et al., 2015; Moosavi-Dezfooli et al., 2016), invisible to human observers, resulting in significant variations in the network's outputs (Antun et al., 2020).Various strategies to improve the robustness of PD-DL networks have been proposed to counter adversarial attacks in MRI reconstruction (Cheng et al., 2020; Calivá et al., 2021; Jia et al., 2022; Raj et al., 2020; Liang et al., 2023). However, all these methods require retraining of the network, incurring a high computational cost, while also having a tendency to lead to additional artifacts for clean/non-attack inputs (Tsipras et al., 2019).

In this work, we propose a novel mitigation strategy for adversarial attacks on DL-based MRI reconstruction, which does not require *any retraining*. Our approach utilizes the idea of cyclic measurement consistency (Zhao & Hu, 2008; Kim et al., 2023; Tachella et al., 2022; Zhang et al., 2024) with

synthesized undersampling patterns. The overarching idea for cyclic measurement consistency is to simulate new measurements from inference results with a new forward model that is from a similar distribution as the original forward model, thus consistent with the original inference. This idea has been used to improve parallel imaging (Zhao & Hu, 2008), then rediscovered in the context of DL reconstruction training (Kim et al., 2023; Tachella et al., 2022; Zhang et al., 2024) and uncertainty guidance (Zhang & Akçakaya, 2024). In our work, we use this idea in a completely novel direction to characterize and mitigate adversarial attacks. Succinctly, without an attack, reconstructions on synthesized measurements should be cycle-consistent, while with a small adversarial perturbation, there should be large discrepancies between reconstructions from actual versus synthesized measurements. We use this consistency to devise an objective function over the network input to effectively mitigate adversarial perturbations. Our contributions are as follows:

- We propose a novel mitigation strategy for adversarial attacks, which optimizes cyclic measurement consistency over the input within a small ball without requiring *any retraining*.
- We show that the mitigation strategy can be applied in a manner that is blind to the size of the perturbation or the algorithm that was used to generate the attack.
- For the first time, we provide a *realistic* scenario for small adversarial attacks in MRI reconstruction, related to impulse noise in k-space, associated with herringbone artifacts (Stadler et al., 2007), as a sparse & bounded adversarial attack. We show our method also mitigates such attacks.
- Our method readily combines with existing robust training strategies to further improve reconstruction quality of DL-based MRI reconstruction under adversarial attacks.
- Our results demonstrate effectiveness across various datasets, PD-DL networks, attack types and strengths, and undersampling patterns, outperforming existing methods qualitatively and quantitatively, without affecting the performance on non-perturbed images.
- Finally, we show that the physics-driven nature of our method makes it robust even to adaptive attacks, where the attacker is aware of the defense strategy and finds the worst-case perturbation that maximize its effectiveness in bypassing the defense algorithm.

## 2 BACKGROUND AND RELATED WORK

### 2.1 PD-DL RECONSTRUCTION FOR ACCELERATED MRI

In MRI, raw measurements are collected in the frequency domain, known as the k-space, using multiple receiver coils, where each coil is sensitive to different parts of the field-of-view. Accelerated MRI techniques acquire sub-sampled data, $\mathbf{y}_\Omega = \mathbf{E}_\Omega \mathbf{x} + \mathbf{n}$, where $\mathbf{E}_\Omega \in \mathbb{C}^{M \times N}$ is the forward multi-coil encoding operator, with $M > N$ in the multi-coil setup (Pruessmann et al., 1999), $\Omega$ is the undersampling pattern with acceleration rate $R$, $\mathbf{n}$ is measurement noise, and $\mathbf{x}$ is the image to be reconstructed. The inverse problem for this acquisition model is formulated as

$$\arg\min_{\mathbf{x}} \|\mathbf{y}_\Omega - \mathbf{E}_\Omega \mathbf{x}\|_2^2 + \mathcal{R}(\mathbf{x}) \tag{1}$$

where the first quadratic term enforces data fidelity (DF) with the measurements, while the second term is a regularizer, $\mathcal{R}(\cdot)$. The objective in Eq. (1) is conventionally solved using iterative algorithms (Fessler, 2020) that alternate between DF and a model-based regularization term.

On the other hand, PD-DL commonly employs a technique called algorithm unrolling (Monga et al., 2021), which unfolds such an iterative reconstruction algorithm for a fixed number of steps. Here, the DF is implemented using conventional methods with a learnable parameter, while the proximal operator for the regularizer is implemented implicitly by a neural network (Hammernik et al., 2023). The unrolled network is trained end-to-end in a supervised manner using fully-sampled reference data (Hammernik et al., 2018; Aggarwal et al., 2019) using a loss of the form:

$$\arg\min_{\boldsymbol{\theta}} \mathbb{E}\Big[\mathcal{L}\big(f(\mathbf{z}_\Omega, \mathbf{E}_\Omega; \boldsymbol{\theta}), \mathbf{x}_{\text{ref}}\big)\Big], \tag{2}$$

where $\mathbf{z}_\Omega = \mathbf{E}_\Omega^H \mathbf{y}_\Omega$ is the zerofilled image that is input to the PD-DL network; $f(\cdot, \cdot; \boldsymbol{\theta})$ is the output of the PD-DL network, parameterized by $\boldsymbol{\theta}$, in image domain; $\mathcal{L}(\cdot, \cdot)$ is a loss function; $\mathbf{x}_{\text{ref}}$ is the reference image. In this work, we unroll the variable splitting with quadratic penalty algorithm (Fessler, 2020), as in MoDL (Aggarwal et al., 2019).

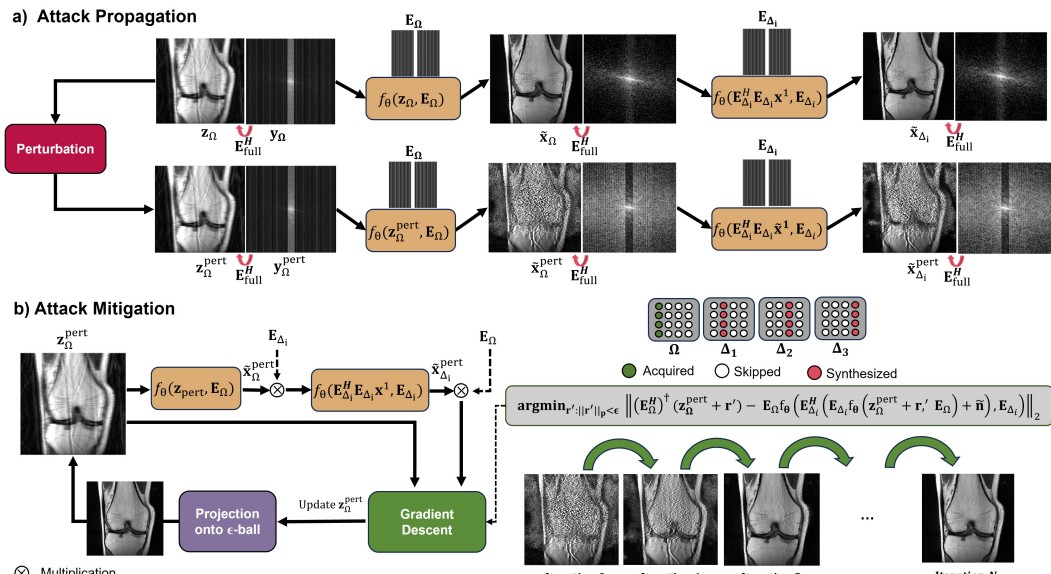

Figure 1: Overview of the proposed mitigation strategy. a) If there is an adversarial attack, the k-space corresponding to the reconstructions of MRI data synthesized from previous DL model outputs will be disrupted. b) This idea is used to devise a novel loss function to find a "corrective" perturbation around the input that ensures cyclic measurement consistency.

## 2.2 ADVERSARIAL ATTACKS IN PD-DL MRI RECONSTRUCTION

Adversarial attacks create serious challenges for PD-DL MRI reconstruction, where small, visually imperceptible changes to input data can lead to large errors in the reconstructed image (Zhang et al., 2021; Antun et al., 2020; Calivá et al., 2021). These find the worst-case degradation $\mathbf{r}$ within a small $\ell_p$ ball that will lead to the largest perturbation in the output of the network (Antun et al., 2020):

$$\arg \max_{\mathbf{r}:||\mathbf{r}||_p \leq \epsilon} \mathcal{L}\big(f(\mathbf{z}_\Omega + \mathbf{r}, \mathbf{E}_\Omega; \boldsymbol{\theta}), f(\mathbf{z}_\Omega, \mathbf{E}_\Omega; \boldsymbol{\theta})\big). \tag{3}$$

We note that this attack calculation is unsupervised, which is the relevant scenario for MRI reconstruction (Jia et al., 2022; Zhang et al., 2021), as the attacker cannot know the fully-sampled reference for a given undersampled dataset. In MRI reconstruction, $\ell_\infty$ perturbations are commonly used in image domain (Zhang et al., 2021; Antun et al., 2020; Liang et al., 2023; Jia et al., 2022), while $\ell_2$ perturbations are used in k-space (Raj et al., 2020) due to scaling differences between low and high-frequency in Fourier domain. In this work, we concentrate on the former, while examples for the latter are provided in Appendix D.7. We also note that image domain attacks can be converted to k-space as: $\mathbf{w} = (\mathbf{E}_\Omega^H)^\dagger \mathbf{r} = \mathbf{E}_\Omega (\mathbf{E}_\Omega^H \mathbf{E}_\Omega)^{-1} \mathbf{r}$, since $M > N$ for multi-coil MRI acquisitions (Pruessmann et al., 1999). Note $\mathbf{w}$ is only non-zero at $\Omega$, and its zerofilled image is $\mathbf{E}_\Omega^H \mathbf{w} = \mathbf{r}$, as expected. In other words, $\ell_\infty$ attacks have k-space representations, where only the acquired locations $\Omega$ are perturbed, aligning with the underlying physics of the problem.

Adversarial attacks are typically calculated using a gradient-based strategy (Goodfellow et al., 2015; Mkadry et al., 2018), where the input is perturbed in the direction of maximal change within the $\ell_\infty$ ball. In this study, we use iterative projected gradient descent (PGD) (Mkadry et al., 2018), as it leads to more drastic perturbations than the single-step fast gradient sign method (FGSM) (Goodfellow et al., 2015). Further results with FGSM are included in Appendix D.6. Finally, we note that neural network based attacks have also been used (Raj et al., 2020), but these are mainly preferred for reduced computation time in training, and often fail to match the degradation caused by iterative optimization-based techniques (Jaeckle & Kumar, 2021).

## 2.3 DEFENSE AGAINST ADVERSARIAL ATTACKS IN MRI RECONSTRUCTION

Adversarial training (AT) incorporates an adversarial term in the training objective for robust training, and has been used both in the image domain (Jia et al., 2022) or k-space (Raj et al., 2020). The

two common approaches either enforce perturbed outputs to the reference (Jia et al., 2022):

$$\min_{\boldsymbol{\theta}} \ \mathbb{E}\Big[ \max_{\|\mathbf{r}\|_\infty \leq \epsilon} \mathcal{L}[f_{\boldsymbol{\theta}}(\mathbf{z}_\Omega + \mathbf{r}, \mathbf{E}_\Omega; \boldsymbol{\theta}), \mathbf{x}_{\text{ref}})]\Big] \tag{4}$$

or aim to balance normal and perturbed training (Raj et al., 2020):

$$\min_{\boldsymbol{\theta}} \ \mathbb{E}\Big[ \max_{\|\mathbf{r}\|_\infty \leq \epsilon} \mathcal{L}[f_{\boldsymbol{\theta}}(\mathbf{z}_\Omega, \mathbf{E}_\Omega; \boldsymbol{\theta}), \mathbf{x}_{\text{ref}})] + \lambda \mathcal{L}[f_{\boldsymbol{\theta}}(\mathbf{z}_\Omega + \mathbf{r}, \mathbf{E}_\Omega; \boldsymbol{\theta}), \mathbf{x}_{\text{ref}})]\Big], \tag{5}$$

where $\lambda$ is a hyperparameter controlling the trade-off. While such training strategies improve robustness against adversarial attacks, it often comes at the cost of reduced performance on non-perturbed inputs (Tsipras et al., 2019). Another recent method for robust PD-DL reconstruction proposes the idea of smooth unrolling (SMUG) (Liang et al., 2023). SMUG (Liang et al., 2023) modifies denoised smoothing (Salman et al., 2020), introduces robustness to a regularizer part of the unrolled network. Each unrolled unit of SMUG performs:

$$\mathbf{x}_s^{(i+1)} = \arg\min_{\mathbf{x}} \|\mathbf{E}_\Omega \mathbf{x}_s^{(i)} - \mathbf{y}_\Omega\|_2^2 + \lambda\|\mathbf{x} - \mathbb{E}_\eta[\mathcal{D}_{\boldsymbol{\theta}}(\boldsymbol{x}_s^{(i)} + \boldsymbol{\eta})]\|_2^2 \tag{6}$$

where $\mathcal{D}_{\boldsymbol{\theta}}$ represents the denoiser network with parameters $\boldsymbol{\theta}$, and $\boldsymbol{\eta} \sim \mathcal{N}(\mathbf{0}, \sigma^2\mathbf{I})$ is random Gaussian noise. During the training, SMUG (Liang et al., 2023) incorporates $P$ Monte Carlo sampling to smooth the denoiser outputs, averaging them before entering the next DF block.

### 2.4 WHY ARE ADVERSARIAL ATTACKS IMPORTANT IN DL MRI RECONSTRUCTION?

**Non-zero probability of worst-case perturbations.** MRI reconstruction pipelines are closed proprietary systems (Winter et al., 2024), thus it is unlikely that an adversary may successfully inject adversarial perturbations during this process. Nonetheless, adversarial attacks provide a *controlled* means to understand the worst-case stability and overall robustness of DL-based reconstruction systems (Antun et al., 2020; Gottschling et al., 2025; Zhang et al., 2021; Han et al., 2024; Alkhouri et al., 2024). It has been argued both empirically (Antun et al., 2020) and theoretically (Gottschling et al., 2025) that worst-case perturbations are not rare events. In particular, if one samples a new input from a small ball around the worst-case perturbation this still leads to a failed reconstruction (Antun et al., 2020). Recent work further shows that sampling from Gaussian noise, i.e. the thermal noise model in MRI, leads to such an instability with non-zero probability (Gottschling et al., 2025).

**Connection to herringbone artifacts.** Apart from Gaussian noise, there are several other causes of perturbations in an MRI scan, including body motion (Zaitsev et al., 2015) or hardware issues (Kashani et al., 2020), which are hard to model mathematically in general, but whose combined effect may lead to similar instabilities for DL-based reconstruction (Antun et al., 2020). One such hardware-related issue is electromagnetic spikes from the gradient power fluctuation or inadequate room shielding, resulting in impulse noise in k-space, which manifest as herringbone artifacts in image domain (Stadler et al., 2007; Jin et al., 2017). When the impulse intensities are high, these artifacts are visible even in fully-sampled data. However, lower intensity impulses may adversely affect DL reconstruction. For the first time, we show this using a sparse and bounded attack model.

**Understanding broader perturbations.** Adversarial perturbations and mitigation algorithms, like ours, are critical to understand the robustness of DL reconstruction models in important scenarios, such as performance for rare pathologies (Muckley et al., 2021). However, these physiological changes are much harder to model and simulate, unlike adversarial attacks, which provide insights into worst-case stability. Finally, we note that our mitigation algorithm is also applicable to unrolled networks in general, and may have applications in broader computational imaging scenarios.

## 3 PROPOSED METHOD FOR TRAINING-FREE MITIGATION OF ADVERSARIAL ATTACKS IN PD-DL MRI

### 3.1 ATTACK PROPAGATION IN SIMULATED K-SPACE

The idea behind our mitigation strategy stems from cyclic measurement consistency with synthesized undersampling patterns, which has been previously used to improve calibration/training of

MRI reconstruction models (Zhao & Hu, 2008; Kim et al., 2023; Tachella et al., 2022; Zhang & Akçakaya, 2024; Zhang et al., 2024). For reconstruction purposes, a well-trained model should generalize to undersampling patterns with similar distributions as the acquisition one (Knoll et al., 2020a). To this end, let $\{\Delta_n\}$ be undersampling patterns drawn from a similar distribution as $\Omega$, including same acceleration rate, similar underlying distribution, e.g. variable density random, and same number of central lines. Further let

$$\tilde{\mathbf{x}}_\Omega = f(\mathbf{z}_\Omega, \mathbf{E}_\Omega; \boldsymbol{\theta}) \tag{7}$$

be the reconstruction of the acquired data. We simulate new measurements $\tilde{\mathbf{y}}_{\Delta_i}$ from $\tilde{\mathbf{x}}$ using the encoding operator $\mathbf{E}_{\Delta_n}$ with the same coil sensitivity profiles as $\mathbf{E}_\Omega$, and let $\mathbf{z}_{\Delta_i} = \mathbf{E}_{\Delta_i}^H \tilde{\mathbf{y}}_{\Delta_i}$ be the corresponding zerofilled image. Then the subsequent reconstruction

$$\tilde{\mathbf{x}}_{\Delta_i} = f(\mathbf{z}_{\Delta_i}, \mathbf{E}_{\Delta_i}; \boldsymbol{\theta}) \tag{8}$$

should be similar to $\tilde{\mathbf{x}}_\Omega$. In particular, we evaluate the similarity over the acquired k-space locations, $\Omega$, as we will discuss in Section 3.2. However, if there is an attack on the acquired lines, either generated directly in k-space or in image domain as discussed in Section 2.2, then this consistency with synthesized measurements are no longer expected to hold, as illustrated in Fig. 1a.

This can be understood in terms of what the PD-DL network does during reconstruction as it alternates between DF and regularization. The DF operation will ensure that the network is consistent with the input measurements, $\mathbf{y}_\Omega$, or equivalently the zerofilled image, $\mathbf{z}_\Omega$. If there is no adversarial attack, we expect the output of a well-trained PD-DL network to be consistent with these measurements, while also showing no sudden changes in k-space (Knoll et al., 2020a). On the other hand, if there is an attack, the output will still be consistent with the measurements, as the attack is designed to be a small perturbation on $\mathbf{y}_\Omega$ or $\mathbf{z}_\Omega$, and thus the small changes on these lines will be imperceptible. Instead, the attack will affect all the other k-space locations $\Omega^C$, the complement of the acquired index set, leading to major changes in these lines for the output of the PD-DL network, as depicted in Fig. 1a. Thus, when we resample a new set of indices $\Delta_i$ that includes lines from $\Omega^C$, under attack the next level reconstruction $\tilde{\mathbf{x}}_{\Delta_i}$ will no longer be consistent with the original k-space data $\mathbf{y}_\Omega$, as measured through $||\mathbf{y}_\Omega - \mathbf{E}_\Omega \tilde{\mathbf{x}}_{\Delta_i}||_2$. The distortion in the k-space will further propagate as we synthesize more levels of data and reconstruct these, if there is an adversarial attack. The following theorem further confirms this intuition:

**Theorem 1.** *Let $\mathbf{y}_\Omega$ and $\tilde{\mathbf{y}}_\Omega = \mathbf{y}_\Omega + \mathbf{w}$ be the clean and perturbed measurements, respectively, and let $\mathbf{x}$ and $\tilde{\mathbf{x}}$ denote the corresponding outputs of the PD-DL network. Then*

$$\|\mathbf{E}_\Omega(\tilde{\mathbf{x}} - \mathbf{x})\|_2 \leq C\|\mathbf{w}\|_2, \tag{9}$$

*where $C$ is a function of the smallest and largest singular values of $\mathbf{E}_\Omega$ and $\mathbf{E}_{\Omega^C}$, a constant characterizing the high-frequency energy of the smooth coil sensitivity maps, the learned DF penalty parameter in MoDL, the number of unrolls, and the Lipschitz constant of the proximal network.*

Proof and details on the constants, are given in Appendix G. Since $\|\tilde{\mathbf{x}} - \mathbf{x}\|_2^2 = \|\mathbf{E}_\Omega(\tilde{\mathbf{x}} - \mathbf{x})\|_2^2 + \|\mathbf{E}_{\Omega^C}(\tilde{\mathbf{x}} - \mathbf{x})\|_2^2$, the theorem implies the residual error on the complementary set $\Omega^C$ must be large.

This description of the attack propagation suggests a methodology for detecting such attacks; however, this is not the focus of this paper. As discussed in Appendix D.4, mitigation can be applied on all inputs, regardless of whether they have been attacked, as the algorithm does not degrade the reconstruction if the input is unperturbed. Thus, to keep the exposition clearer, we focus on mitigation for the reminder of the paper, and a threshold-based detection scheme is discussed in Appendix C.

## 3.2 ATTACK MITIGATION WITH CYCLIC CONSISTENCY

Based on the characterization of the attack propagation, we next introduce our proposed training-free mitigation strategy. We note that adversarial attacks of Section 2.2 all aim to create a small perturbation within a ball around the original input. Here the size of the ball specifies the attack strength, the particular algorithm specifies how the attack is generated within the given ball, and the attack domain/norm specifies the type of $\ell_p$ ball and whether it is in k-space or image domain.

Succinctly, our mitigation approach aims to reverse the attack generation process, by searching within a small ball around the perturbed input to find a clear input. The objective function for this

task uses the aforementioned idea of cyclic measurement consistency, and is given as

$$\arg\min_{\mathbf{r}':\|\mathbf{r}'\|_p \leq \epsilon} \mathbb{E}_\Delta\left[\left\|(\mathbf{E}_\Omega^H)^\dagger(\mathbf{z}_\Omega + \mathbf{r}') - \mathbf{E}_\Omega f\Big(\mathbf{E}_\Delta^H\big(\mathbf{E}_\Delta f(\mathbf{z}_\Omega + \mathbf{r}', \mathbf{E}_\Omega; \boldsymbol{\theta}) + \tilde{\mathbf{n}}\big), \mathbf{E}_\Delta; \boldsymbol{\theta}\Big)\right\|_2\right], \quad (10)$$

where $\mathbf{r}'$ is a small "corrective" perturbation and $\mathbf{z}_\Omega + \mathbf{r}'$ corresponds to the mitigated/corrected input. The first term, $(\mathbf{E}_\Omega^H)^\dagger(\mathbf{z}_\Omega + \mathbf{r}')$ is the minimum $\ell_2$ k-space solution that maps to this zerofilled image (Zhang et al., 2021). The second term is the corresponding k-space values at the acquired indices $\Omega$ after two stages of cyclic reconstruction. Note a small noise term, $\tilde{\mathbf{n}}$, is added to the synthesized data to maintain similar signal-to-noise-ratio (Zhang et al., 2024; Knoll et al., 2019). The expectation is taken over undersampling patterns $\Delta$ with a similar distribution to the original pattern $\Omega$.

The objective function is solved using a reverse PGD approach, as detailed in Algorithm 1. Note the algorithm performs the expectation in Eq. (10) over $K$ sampling pattern $\{\Delta_k\}_{k=1}^K$. Notably, our reverse PGD performs a gradient descent instead of the ascent in PGD (Mkadry et al., 2018), and includes a projection on to the $\epsilon$ ball to ensure the solution remains within the desired neighborhood.

Finally, this algorithm uses the strength of the attack, but it is practically beneficial to mitigate the attack without it, as this will not always be available to the end user. In this blind setup, we additionally optimize its input parameters $\epsilon$ and $\alpha$ jointly with Eq. (10) in an iterative manner. First, we decrease $\epsilon$ with a linear scheduler for a fixed $\alpha$, starting from a large ball until convergence. Subsequently, we fix $\epsilon$ and decrease $\alpha$ similarly. The alternating process can be repeated, though in practice, one stage is sufficient. Finally, for blind mitigation, we always use $\ell_\infty$ ball, even for $\ell_2$ attacks in k-space discussed in Appendix E, as it contains the $\ell_2$ ball of the same radius.

---

**Algorithm 1** Attack Mitigation

**Require:** $\epsilon, \alpha, \mathbf{z}_\Omega^{pert}, \mathbf{E}_\Omega, \{\mathbf{E}_{\Delta_k}\}_{k=1}^K, f(\cdot, \cdot; \boldsymbol{\theta})$     ▷ Inputs
**Ensure:** Clean version of $\mathbf{z}_\Omega^{pert}$    ▷ Mitigate attack on input
1:   $\tilde{\mathbf{z}}_\Omega = \mathbf{z}_\Omega^{pert}$
2: **repeat**
3:    Loss $= 0$
4:    **for** $k = 1$ to $K$ **do**
5:      $\tilde{\mathbf{y}}_\Omega = (\mathbf{E}_\Omega^H)^\dagger \tilde{\mathbf{z}}_\Omega$
6:      $\tilde{\tilde{\mathbf{y}}}_\Omega = \mathbf{E}_\Omega f\big(\mathbf{E}_{\Delta_k}^H(\mathbf{E}_{\Delta_k} f(\tilde{\mathbf{z}}_\Omega, \mathbf{E}_\Omega; \boldsymbol{\theta}) + \tilde{\mathbf{n}}), \mathbf{E}_{\Delta_k}; \boldsymbol{\theta}\big)$
7:      $\text{loss}_k = \|\tilde{\mathbf{y}}_\Omega - \tilde{\tilde{\mathbf{y}}}_\Omega\|_2$     ▷ Eq. 10
8:      Loss $=$ Loss $+ \text{loss}_k$
9:    **end for**
10:   $\mathbf{grad} = \frac{1}{K}\nabla_{\tilde{\mathbf{z}}_\Omega}$Loss
11:   $\tilde{\mathbf{z}}_\Omega = \tilde{\mathbf{z}}_\Omega - \alpha \cdot \text{sgn}(\mathbf{grad})$
12:   $\tilde{\mathbf{z}}_\Omega = \text{clip}_{\mathbf{z}_\Omega^{pert}, \epsilon}(\tilde{\mathbf{z}}_\Omega)$     ▷ Projection to $\epsilon$ ball
13: **until** Converge

---

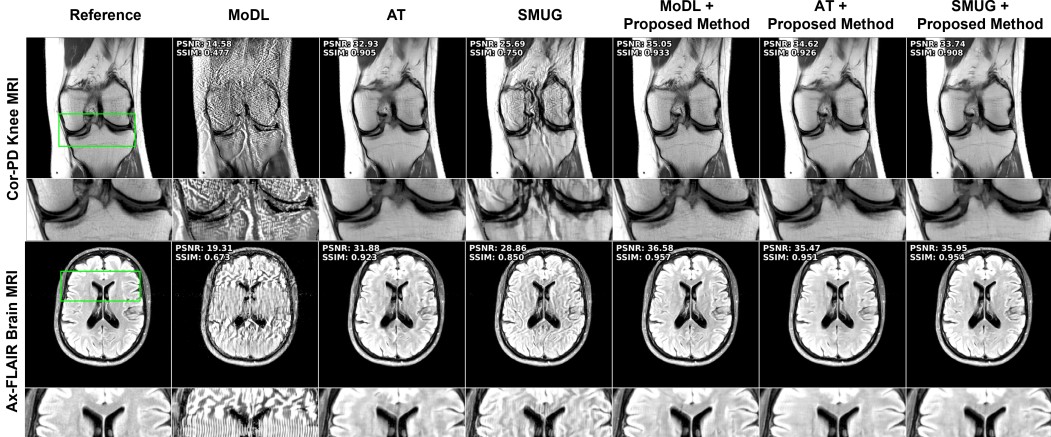

Figure 2: Representative reconstruction results for Cor-PD knee, and Ax-FLAIR brain MRI Datasets at $R = 4$. The attack inputs lead to severe disruption in the baseline MoDL reconstruction. Adversarial training improves these, albeit suffering from blurriness. SMUG fails to eliminate the attack. The proposed strategy reduces the artifacts and maintains sharpness. Furthermore it can be combined with the other strategies for further gains (last two columns).

### 3.3 MITIGATION PERFORMANCE ON ADAPTIVE ATTACKS

Recent works suggest that a good performance on iterative optimization-based attacks may not be a good indicator of robustness, as the class of adaptive attacks can jointly deceive the baseline (reconstruction) network and bypass the defense once the attacker is aware of the defense strategy (Carlini & Wagner, 2017). Consequently, they have become the standard when evaluating defenses (Tramer et al., 2020). To generate adaptive attacks, our mitigation in Algorithm 1 needs to be incorporated into the attack generation objective Eq. (3). To simplify the notation, we define our mitigation function based on Eq. (10) as

$$g(\mathbf{z}_\Omega) \triangleq \min_{\mathbf{r}':||\mathbf{r}'||_p \leq \epsilon} \mathbb{E}_\Delta \left[ \left\| (\mathbf{E}_\Omega^H)^\dagger (\mathbf{z}_\Omega + \mathbf{r}') - \mathbf{E}_\Omega f\left( \mathbf{E}_\Delta^H \left( \mathbf{E}_\Delta f(\mathbf{z}_\Omega + \mathbf{r}', \mathbf{E}_\Omega; \boldsymbol{\theta}) + \tilde{\mathbf{n}} \right), \mathbf{E}_\Delta; \boldsymbol{\theta} \right) \right\|_2 \right]$$

which leads to the adaptive attack generation objective:

$$\arg \max_{\mathbf{r}:||\mathbf{r}||_p \leq \epsilon} \mathcal{L}\left( f(\mathbf{z}_\Omega + \mathbf{r}, \mathbf{E}_\Omega; \boldsymbol{\theta}), f(\mathbf{z}_\Omega, \mathbf{E}_\Omega; \boldsymbol{\theta}) \right) + \lambda \, g(\mathbf{z}_\Omega + \mathbf{r}), \tag{11}$$

where the first term finds a perturbation that fools the baseline, as in Eq. (3), while the second term integrates our mitigation. Thus, maximizing Eq. (11) leads to a perturbation $\mathbf{r}$ that not only misleads the baseline reconstruction, but also maximizes the mitigation loss, resulting in an adaptive attack.

## 4 EXPERIMENTS

### 4.1 IMPLEMENTATION DETAILS

**Datasets.** Multi-coil coronal proton density (Cor-PD) knee and axial FLAIR (Ax-FLAIR) brain MRI from fastMRI database (Knoll et al., 2020b), respectively with 15 and 20 coils, were used. Retrospective equispaced undersampling was applied at acceleration $R = 4$ to the fully-sampled data with 24 central auto-calibrated signal (ACS) lines.

**Baseline Network.** The PD-DL network used in this study was a modified version of MoDL (Aggarwal et al., 2019), unrolled for 10 steps, where a ResNet regularizer was used (Yaman et al., 2022b). Further details about the architecture and training are provided in Appendix A. All comparison methods were implemented using this MoDL network to ensure a fair comparison, except for the results on the applicability of our method to different PD-DL networks.

**Attack Generation Details.** PGD (Mkadry et al., 2018) was used to generate the attacks in an unsupervised manner, as detailed earlier for a realistic setup. Additional results with supervised attacks and FGSM are provided in Appendix D.5 and D.6, respectively, and lead to the same conclusions. Complex images were employed to generate the attack and gradients, and MSE loss was used.

**Comparison Methods.** We compared our mitigation approach with existing robust training methods, including adversarial training (Jia et al., 2022; Raj et al., 2020) and Smooth Unrolling (SMUG) (Liang et al., 2023). Adversarial training was implemented using Eq. (4) (Jia et al., 2022), while results using Eq. (5) (Raj et al., 2020) is provided in Appendix D.3. Further implementation details for all methods are provided in Appendix A.

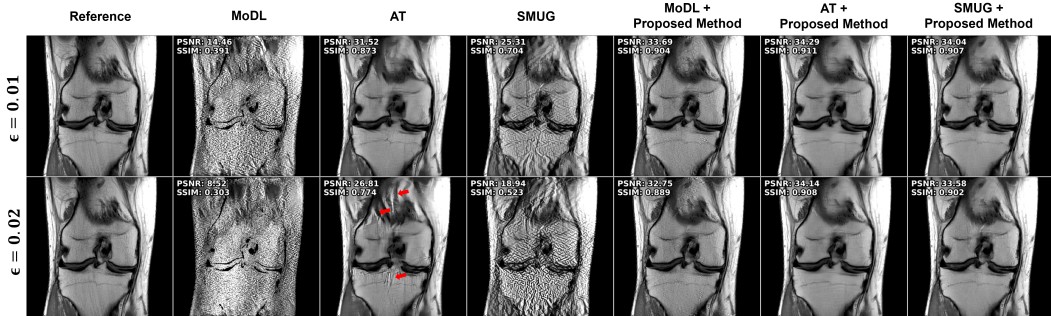

Figure 3: Performance across different attack strengths. Both Adversarial Training and SMUG fail to perform well against attack strengths they were not trained on. In contrast, the proposed training-free mitigation shows good performance across perturbation levels.

**Cyclic Consistency Details.** The synthesized masks $\{\Delta_k\}$ were generated by shifting the equispaced undersampling patterns by one line while preserving the ACS lines (Zhang et al., 2024). In this setting, the number of synthesized masks is $R - 1$.

**Adaptive Attack Details.** Direct optimization of Eq. (11) requires the solution of a long computation graph and multiple nested iterations of neural networks. However, this may induce gradient obfuscation, leading to a false sense of defense security (Athalye et al., 2018). Thus, we followed the gradient computation strategy of (Chen et al., 2023), by unrolling $g(\cdot)$ in Eq. (11) first (Yang et al., 2022), and then backpropagating through the whole objective. Hence, we let $g_T(\cdot)$ be the $T$-step unrolled version of $g(\cdot)$, and report performance for different $T$. Additional information about checkpointing for large $T$, tuning of $\lambda$ and noise precalculation for $\tilde{\mathbf{n}}$ are provided in Appendix F.

## 4.2 ATTACK MITIGATION RESULTS

**Performance Across Datasets.** We first study our approach and the comparison methods on knee and brain MRI datasets at $R = 4$. Fig. 2 shows that baseline PD-DL (MoDL) has substantial artifacts under attack. SMUG improves these but still suffers from noticeable artifacts. AT resolves the artifacts, albeit with blurring. Our proposed approach successfully mitigates the attacks *without any retraining*, while maintaining sharpness. We note our method can also be combined with SMUG and AT to further improve performance. Tab. 1 summarizes the quantitative metrics for all test slices, consistent with visual observations.

Table 1: SSIM/PSNR on all test slices.

| Dataset | Metric | SMUG | Adversarial Training (AT) | Proposed Method + MoDL / SMUG / AT |
|---|---|---|---|---|
| Cor-PD | PSNR | 28.22 | 33.99 | 35.14 / 34.85 / **36.57** |
| | SSIM | 0.79 | 0.92 | 0.92 / 0.92 / **0.94** |
| Ax-FLAIR | PSNR | 29.67 | 34.03 | **36.41** / 34.67 / 35.63 |
| | SSIM | 0.84 | 0.91 | **0.95** / 0.92 / 0.94 |

**Performance Across Attack Strengths and Blind Mitigation**. We next test the methods across different attack strengths, $\epsilon \in \{0.01, 0.02\}$. Fig. 3 shows the results for both attack strengths using the robust training methods trained with $\epsilon = 0.01$ and proposed mitigation. As in Fig. 2, SMUG has artifacts at $\epsilon = 0.01$, which gets worse at $\epsilon = 0.02$. Similarly, AT struggles at $\epsilon = 0.02$, since it was trained at $\epsilon = 0.01$, leading to visible artifacts (arrows). On the other hand, our training-free mitigation is successful at both $\epsilon$. This is expected, since no matter how big the $\epsilon$ ball is, our mitigation explores the corresponding vicinity of the perturbed sample to optimize Eq. (10). Further quantitative results are in Appendix D.1. Implementation details and results on blind mitigation without knowledge of attack type/strength are in Appendix E.

**Performance Across Different PD-DL Networks.** Next, we hypothesize that our method is agnostic to the PD-DL architecture. To test this hypothesis, we perform our mitigation approach for different unrolled networks, including XPDNet (Ramzi et al., 2020), Recurrent Inference Ma-

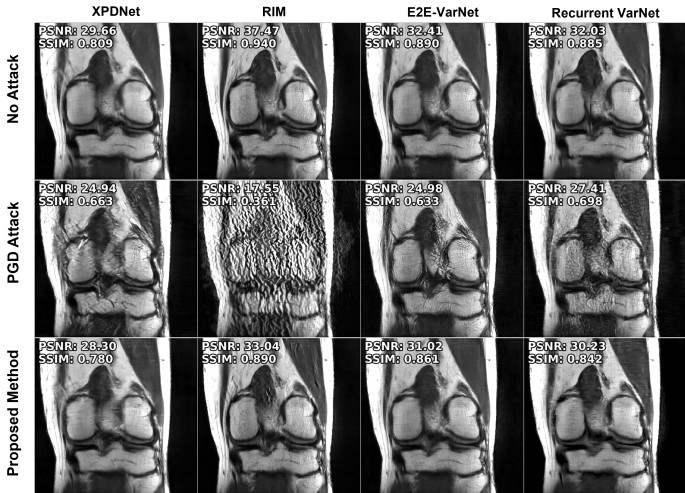

Figure 4: Proposed mitigation approach is readily applicable to various PD-DL networks for MRI reconstruction.

chine (Lønning et al., 2019), E2E-VarNet (Sriram et al., 2020), and Recurrent-VarNet (Yiasemis et al., 2022b). The implementation details are discussed in Appendix A.1. Fig. 4 depicts representative images for clear and perturbed inputs, and our proposed cyclic mitigation results. Overall, all networks show artifacts for perturbed inputs, while our proposed cyclic mitigation algorithm works well on all of them to reduce these artifacts. Further quantitative metrics for these networks are in Appendix D.2.

**Performance Against Herringbone Artifacts.** Next, we assess the mitigation algorithm against an $\ell_0$ attack, simulating small spikes in k-space, which may occur due to hardware issues (Stadler et al., 2007). Fig. 5 shows how a few small spikes can lead to instabilities in MoDL, similar to herringbone artifacts. Our mitigation effectively removes these. Further implementation details and quantitative results are provided in the Appendix B.

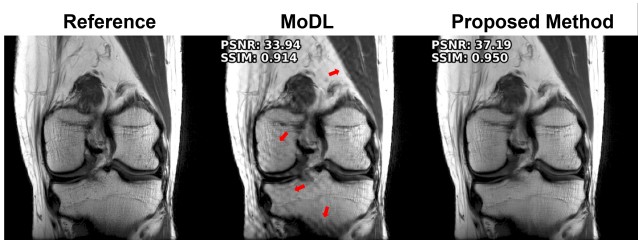

Figure 5: Mitigation of a herringbone ($\ell_0$) perturbation.

**Performance Against Adaptive Attacks.** Tab. 2 shows the performance of our mitigation algorithm for adaptive attacks with $T \in \{10, 25, 50, 100\}$ unrolls. Due to the high computational cost of generating adaptive attacks for $T = 100$, we ran the adaptive attack mitigation on a subset of 75 Cor-PD slices, which is why the non-adaptive attack results have lower PSNR

Table 2: PSNR for adaptive attacks on 75 Cor-PD slices. Parentheses in the last column indicate the mean iteration for convergence of the iterative algorithm.

| Attack Type | #Unrolls (T) | Baseline Reconstruction | Unrolled Algorithm 1 | Iterative Algorithm 1 |
|---|---|---|---|---|
| Non-adaptive | N/A | 16.16 | N/A | 34.69 |
| Adaptive | 10 | 19.23 | 29.47 | 34.34 (119 iters) |
| Adaptive | 25 | 19.32 | 32.79 | 34.16 (111 iters) |
| Adaptive | 50 | 19.96 | 33.39 | 34.14 (105 iters) |
| Adaptive | 100 | 21.02 | 33.78 | 34.01 (100 iters) |

than the full test set in Tab. 1. For mitigation of adaptive attacks, we ran both an unrolled version (used to generate the adaptive attack) and an iterative version (ran until convergence) of Algorithm 1. Average number of iterations for the latter are reported in paranthesis in the last column. Further visual examples are in Appendix F. We observe the following: 1) Baseline reconstructions have higher PSNR under adaptive attacks than non-adaptive attacks, as adaptive attacks balance two terms, reducing its focus on purely destroying the reconstruction. This effect increases as $T$ increases, as expected. 2) For few number of unrolls, adaptive attack degrades performance if mitigated with the unrolled version. For $T < 50$, the unrolled mitigation struggles ($\sim$ 5dB degradation for $T = 10$) with the adaptive attack designed for matched number of unrolls. 3) Our mitigation readily resolves adaptive attacks if run until convergence. For large $T \geq 50$, unrolled mitigation also largely resolves adaptive attacks. 4) Even though adaptive attacks with large $T$ lead to a weaker baseline attack, they degrade our mitigation more, even though the overall degradation is slight even at $T = 100$ (.68dB).

These observations all align with the physics-driven design of the mitigation: The PD-DL reconstruction network ends with data fidelity, i.e. the network output is consistent with (perturbed) $\mathbf{y}_\Omega$. Since the attack is a tiny perturbation on data at $\Omega$, it will cause misestimation of lines in $\Omega^C$ instead (as in Fig. 1a and Theorem 1). Our method synthesizes new measurements at $\Delta$ from the latter, and uses it to perform a second reconstruction, which are mapped to $\Omega$ and checked for consistency with $\mathbf{y}_\Omega$. Thus, the only way the mitigation can be fooled is if this cyclic consistency is satisfied, which in turn indicates that the intermediate recon on $\Omega^C$ is good, effectively mitigating the attack.

### 4.3 Ablation Study

We perform an ablation study on how many levels of reconstructions are needed for mitigation. In this case, multiple steps of reconstructions and data synthesis can be used to update the loss function in Eq. (10). Results, given in Appendix H, show that enforcing cyclic consistency with multiple levels degrades performance and requires more computational resources. Hence, using 2-cyclic reconstruction stages is the best choice from both performance and computational perspectives.

### 5 Conclusions

In this study, we proposed a method to mitigate small imperceptible adversarial input perturbations on DL MRI reconstructions, without requiring any retraining. We showed our method is robust across different datasets, networks, attack strengths/types, including the practical herringbone attack. Our method can be combined with existing robust training methods to further enhance their performance. Additionally, our technique can be performed in a blind manner without attack-specific information, such as attack strength or type. Finally, owing to its physics-based design, our method is robust to adaptive attacks, which have emerged as the recent standard for robustness evaluation.

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

# A    IMPLEMENTATION DETAILS

## A.1    PD-DL NETWORK DETAILS

**MoDL** implementation is based on (Aggarwal et al., 2019), unrolling variable splitting with quadratic penalty algorithm (Fessler, 2020) for 10 steps. The proximal operator for the regularizer is implemented with a ResNet (Yaman et al., 2020; Hosseini et al., 2020; Demirel et al., 2023b), and data fidelity term is implemented using conjugate gradient, itself unrolled for 10 iterations (Aggarwal et al., 2019). The ResNet comprises input and output convolutional layers, along with 15 residual blocks. Each residual block has a skip connection and two convolutional blocks with a rectified linear unit in between. At the end of each residual block, there is a constant scaling layer (Timofte et al., 2017), and the weights are shared among different blocks (Aggarwal et al., 2019).

**XPDNet** implementation is based on (Yiasemis et al., 2022a) and follows (Ramzi et al., 2020), which unrolls the primal dual hybrid gradient (PDHG) algorithm (Chambolle & Pock, 2011) for 10 steps. Each step contains k-space and image correction in sequence, where form the data fidelity and regularizer respectively. XPDNet applies the undersampling mask on the subtraction of the intermediate k-space with original measurements in k-space correction step. Image correction/regularizer is implemented using multi-scale wavelet CNN (MWCNN) (Liu et al., 2019) followed by a convolutional layer. Inspired by PDNet (Adler & Öktem, 2018), it uses a modified version of PDHG to utilize a number of optimization parameters instead of just using the previous block's output. 5 primal and 1 dual variables are used during the unrolling process, and the weights are not shared across the blocks.

**RIM** implementation based on (Yiasemis et al., 2022a) as described in (Lønning et al., 2019) unrolls the objective for 16 time steps, where each utilizes a recurrent time step. Each time step takes the previous reconstruction, hidden states and the gradient of negative log-likelihood (as data fidelity term) and outputs the incremental step in image domain to take using a gated recurrent units (GRU) structure (Cho, 2014), where it utilizes depth 1 and 128 hidden channels. Parameters are shared across different recurrent blocks.

**E2E-VarNet** uses the publicly available implementation (Sriram et al., 2020), and like variational networks, implements an unrolled network to solve the regularized least squares objective using gradient descent. The algorithm is unrolled for 12 steps. Each step combines data fidelity with a regularizer. Data fidelity term applies the undersampling mask after subtraction of intermediate k-space from the measurements, while learned regularizer is implemented via U-Net (Zhou et al., 2018), where it uses 4 number of pull layers and 18 number of output channels after first convolution layer. Weights are not shared across blocks.

**Recurrent VarNet** uses the publicly available implementation (Yiasemis et al., 2022b) estimates a least squares variational problem by unrolling with gradient descent for 8 steps. Each iteration is a variational block, comprising data fidelity and regularizer terms. Data fidelity term calculates the difference between current level k-space and the measurements on undersampling locations, where regulizer utilizes gated recurrent units (GRU) structure (Cakır et al., 2017). Each unroll block uses 4 of these GRUs with 128 number of hidden channels for regularizer. Parameters are not shared across different blocks (Yiasemis et al., 2022b).

As described in the main text, all methods were retrained on the respective datasets with supervised learning for maximal performance. Unsupervised training that only use undersampled data (Yaman et al., 2020; 2022a; Akçakaya et al., 2022) can also be used, though this typically does not outperform supervised learning.

## A.2    COMPARISON METHODS AND ALGORITHMIC DETAILS

**SMUG** (Liang et al., 2023) trains the same PD-DL network we used for MoDL using smoothing via Eq. (6). Smoothing is implemented using 10 Monte-Carlo samples (Liang et al., 2023), with a noise level of 0.01, where data is normalized in image domain.

**Adversarial Training (AT)** method also uses the same network structure as MoDL. Here, each adversarial sample is generated with 10 iterations of PGD (Mkadry et al., 2018) with $\epsilon = 0.01$ and $\alpha = \epsilon/5$. Data are normalized to $[0, 1]$ in image domain.

## B  HERRINGBONE ARTIFACT DETAILS

Herringbone artifacts can arise from several factors, including electromagnetic spikes by gradient coils, fluctuating in power supply, and RF pulse dependencies (Stadler et al., 2007). These factors introduce impulse-like spikes in the k-space, and if their intensities are high enough, they manifest as herringbone-like artifacts across the entire image, even for fully-sampled acquisitions (Jin et al., 2017). Here, we hypothesized that DL-based reconstruction of undersampled datasets may be affected by such spikes even if the intensities are not visibly apparent in the zerofilled images or in fully-sampled datasets. The standard spike modeling for herringbone artifacts uses a sum of impulses, as follows:

$$\tilde{\mathbf{y}}_\Omega = \mathbf{y}_\Omega + \sum_{j=1}^{D} \xi_j \boldsymbol{\delta}_{i_j}, \tag{12}$$

where $\boldsymbol{\delta}i_j$ is a delta/impulse on the $i_j^{\text{th}}$ index (i.e. the canonical basis vector $\mathbf{e}_{i_j}$), $\xi_j$ is the strength of the spike on the $i_j^{\text{th}}$ index, and $D$ is the number of spikes. Thus, we use the same model and use a sparse and bounded adversarial attack for DL-based reconstructions. In particular, we randomly select the locations of $\{i_j\}_{j=1}^{D}$ with heavier sampling in low-frequencies to highlight the traditional herringbone-type artifacts visibly. While selecting the high-frequency locations is also feasible, the resulting artifacts appear less sinusoidal. Let $\mathbf{w}_{\text{hb}} = \sum_{j=1}^{D} \xi_j \boldsymbol{\delta}_{i_j}$ with unknown spike strength $\xi_j$, we optimize:

$$\arg \max_{\mathbf{w}_{\text{hb}}:||\mathbf{w}_{\text{hb}}||_\infty \leq \epsilon} \mathcal{L}\big(f(\mathbf{E}_\Omega^H(\mathbf{y}_\Omega + \mathbf{w}_{\text{hb}}), \mathbf{E}_\Omega; \boldsymbol{\theta}), f(\mathbf{E}_\Omega^H \mathbf{y}_\Omega, \mathbf{E}_\Omega; \boldsymbol{\theta})\big). \tag{13}$$

This was solved using PGD (Mkadry et al., 2018) with 10 iterations, $D = 25$, and $\epsilon = 0.06 \cdot \max(\mathbf{y}_\Omega)$, ensuring that the resulting zerofilled image remained visibly identical to the clean zerofilled image.

## C  ATTACK DETECTION USING SIMULATED K-SPACE

The description of the attack propagation suggests a methodology for detecting these attacks. Noting that the process is best understood in terms of consistency with acquired data in k-space, we perform detection in k-space instead of attempting to understand the differences between subsequent reconstruction in image domain, which is not clearly characterized. In particular, we define two stages of k-space errors in terms of $\mathbf{y}_\Omega$ for $\tilde{\mathbf{x}}_\Omega$ and $\tilde{\mathbf{x}}_{\Delta_i}$, which were defined in Eq. (7)-Eq. (8) as follows:

$$\zeta_1 = \frac{||\mathbf{y}_\Omega - \mathbf{E}_\Omega \tilde{\mathbf{x}}_\Omega||_2}{||\mathbf{y}_\Omega||_2}, \; \zeta_2 = \frac{||\mathbf{y}_\Omega - \mathbf{E}_\Omega \tilde{\mathbf{x}}_{\Delta_i}||_2}{||\mathbf{y}_\Omega||_2}. \tag{14}$$

From the previous description $\zeta_1$ is expected to be small with or without attack. However, $\zeta_2$ is expected to be much larger under the attack, while it should be almost at the same level as $\zeta_1$ without an attack. Thus, we check the difference between these two normalized errors, $\zeta_2 - \zeta_1$, and detect an attack if it is greater than a dataset-dependent threshold. The process is depicted in Fig. 6, and summarized in Algorithm 2. Fig. 7 shows how $\zeta_2 - \zeta_1$ changes for knee and brain datasets for both PGD and FGSM attacks on normalized zerofilled images for $\epsilon \in \{0.01, 0.02\}$. It is clear that cases with an attack vs. non-perturbed inputs are separated by a dataset-dependent threshold. Note that given the sensitivity of PD-DL networks to SNR and acceleration rate changes, this dataset dependence is not surprising (Knoll et al., 2019), and can be evaluated offline for a given trained model.

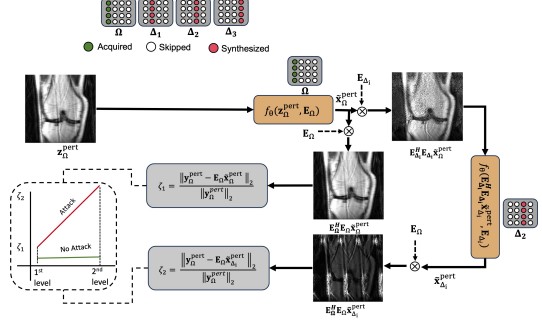

Figure 6: Propagation of the attack in Fig. 1a motivates tracking the normalized $\ell_2$ error on sampled k-space locations after reconstruction; a large change in this error indicates an attack.

# D QUANTITATIVE RESULTS AND REPRESENTATIVE EXAMPLES

Due to space constraints, the figures and results in the main text focused on $\ell_\infty$ attacks generated with unsupervised PGD (Mkadry et al., 2018), as mentioned in Section 4.1. This section provides the corresponding results on related attack types mentioned in Section 4.2.

## D.1 HIGHER ATTACK STRENGTHS

Tab. 3 summarizes the quantitative population metrics for different attack strengths, $\epsilon$, complementing the representative examples shown in Fig. 3 of Section 3.2. These quantitative results align with the visual observations.

## D.2 QUANTITATIVE METRICS FOR DIFFERENT NETWORKS

Tab. 4 shows that the quantitative metrics for the proposed attack mitigation strategy improve substantially compared to the attack for all unrolled networks, aligning with the observations in Fig. 4.

## D.3 DIFFERENT ADVERSARIAL TRAINING METHODS

This subsection provides an alternative implementation of the adversarial training based on Eq. (5) with $\lambda = 1$ to balance the perturbed and clean input, instead of Eq. (4) that was provided in the main text as a comparison. Results in Tab. 5 show that the version in the main text outperforms the alternative version provided here.

## D.4 MITIGATION PERFORMANCE ON NON-PERTURBED DATA

Hence, the mitigation does not degrade the quality of the clean inputs, and does not incur large computational costs, as it effectively converges in a single iteration. Visual examples of this process are depicted in Fig. 8. As discussed in Section 3.1, Algorithm 1 does not compromise the reconstruction quality if the input is unperturbed. This is because, with an unperturbed input image in Eq. (10), the intermediate reconstruction remains consistent with the measurements. As a result, the objective value remains close to zero and stays near that level until the end, indicating the mitigation starts from an almost optimal point of the objective function.

## D.5 SUPERVISED ATTACKS

---

**Algorithm 2** Attack Detection

**Require:** $\mathbf{z}_\Omega, \mathbf{E}_\Omega, \mathbf{E}_\Delta, f(\cdot,\cdot;\boldsymbol{\theta}), \tau$    ▷ Input parameters
**Ensure: True** or **False**, presence of attack    ▷ Output
1: $\tilde{\mathbf{x}}_\Omega \leftarrow f(\mathbf{z}_\Omega, \mathbf{E}_\Omega; \boldsymbol{\theta})$
2: $\mathbf{y}_{\Delta_i} \leftarrow \mathbf{E}_\Delta \tilde{\mathbf{x}}_\Omega + \tilde{\mathbf{n}}$
3: $\tilde{\mathbf{x}}_\Delta \leftarrow f(\mathbf{E}_\Delta^H \mathbf{y}_\Delta, \mathbf{E}_\Delta; \boldsymbol{\theta})$
4: $\zeta_1 = \frac{||\mathbf{y}_\Omega - \mathbf{E}_\Omega \tilde{\mathbf{x}}_\Omega||_2}{||\mathbf{y}_\Omega||_2}$
5: $\zeta_2 = \frac{||\mathbf{y}_\Omega - \mathbf{E}_\Omega \tilde{\mathbf{x}}_\Delta||_2}{||\mathbf{y}_\Omega||_2}$
6: If $\zeta_2 - \zeta_1 \geq \tau$ **True**, else **False**

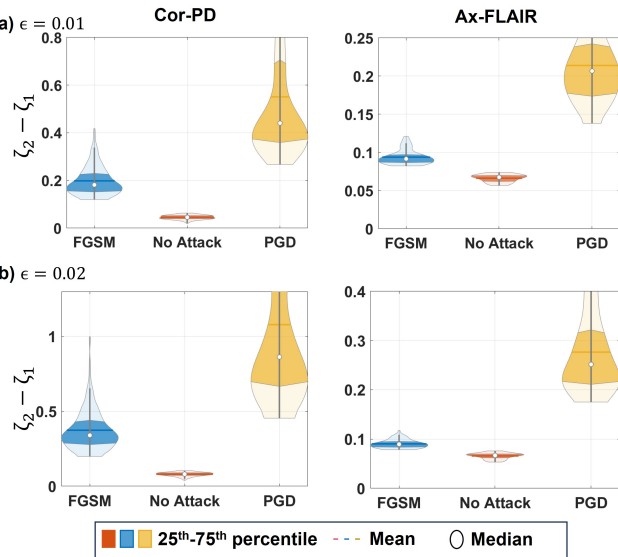

Figure 7: Attack detection for different datasets. $\zeta_2 - \zeta_1$ for different attack types are clearly separated from the no attack case. For stronger attack, $\epsilon = 0.02$, $\zeta_2 - \zeta_1$ is more easily distinguishable. The violin plots show the median and [25,75] percentile in darker colors for easier visualization.

Table 3: Different attack strengths: Quantitative metrics on all test slices of Cor-PD.

| $\epsilon$ | Metric | SMUG | Adversarial Training (AT) | Proposed Method + MoDL / SMUG / AT |
|---|---|---|---|---|
| 0.01 | PSNR | 28.22 | 33.99 | 35.14/34.85/**36.57** |
|  | SSIM | 0.79 | 0.92 | 0.92/0.92/**0.94** |
| 0.02 | PSNR | 21.86 | 30.91 | 33.25/32.97/**33.42** |
|  | SSIM | 0.61 | 0.88 | 0.91/0.91/**0.93** |

| Reference | Iteration 1 | Iteration 5 | Iteration 10 | Iteration 15 |
|:---:|:---:|:---:|:---:|:---:|
| | PSNR: 38.03 SSIM: 0.953 | PSNR: 38.02 SSIM: 0.953 | PSNR: 38.01 SSIM: 0.953 | PSNR: 38.01 SSIM: 0.953 |

Figure 8: Performance of mitigation algorithm on non-perturbed data. The mitigation effectively converges in one iteration. As shown, the algorithm maintains the quality of the clean input.

While Section 4.1 and 4.2 focused on unsupervised attacks due to practicality, here we provide additional experiments with supervised attacks, even though they are not realistic for MRI reconstruction systems. Tab. 6 shows that the proposed method is equally efficient in mitigating supervised attacks.

Table 4: Quantitative metrics for different unrolled networks.

| Network | Metric | With Attack | After Proposed Mitigation |
|:---:|:---:|:---:|:---:|
| XPDNet | PSNR | 25.49 | 29.43 |
| | SSIM | 0.67 | 0.80 |
| RIM | PSNR | 19.63 | 34.81 |
| | SSIM | 0.39 | 0.90 |
| E2E-VarNet | PSNR | 24.24 | 29.52 |
| | SSIM | 0.59 | 0.84 |
| Recurrent VarNet | PSNR | 22.27 | 29.24 |
| | SSIM | 0.52 | 0.84 |

Table 5: Comparison of adversarial training approaches.

| Method | Metric | With Attack |
|:---:|:---:|:---:|
| AT with Eq. (4) | PSNR | 33.99 |
| | SSIM | 0.92 |
| AT with Eq. (5) | PSNR | 33.61 |
| | SSIM | 0.91 |
| AT with Eq. (4) + Proposed Method | PSNR | 36.17 |
| | SSIM | 0.94 |
| AT with Eq. (5) + Proposed Method | PSNR | 36.91 |
| | SSIM | 0.94 |

### D.6 FGSM ATTACK

In Section 4.1, we used the PGD method for attack generation due to the more severe nature of the attacks. Here, we provide additional experiments with FGSM attacks (Goodfellow et al., 2015). Tab. 7 show results using SMUG, adversarial training and our method with FGSM attacks with $\epsilon = 0.01$. Corresponding visual examples are depicted in Fig. 9, showing that all methods perform better under FGSM compared to PGD attacks.

### D.7 $\ell_2$ ATTACKS IN K-SPACE

$\ell_2$ attacks have been used in k-space due to the large variation in intensities in the Fourier domain (Raj et al., 2020). To complement the $\ell_\infty$ attacks in image domain that was provided in the main text, here we provide results for $\ell_2$ attacks in k-space, generated using PGD (Mkadry et al., 2018) for 5 iterations, with $\epsilon = 0.05 \cdot \|\mathbf{y}_\Omega\|_2$ and $\alpha = \frac{\epsilon}{5}$. Fig. 10 depicts representative re-

Table 6: Mitigation with supervised vs. unsupervised attacks.

| Attack Method | Metric | Proposed Method |
|:---:|:---:|:---:|
| Unsupervised Attack | PSNR | 32.44 |
| | SSIM | **0.91** |
| Supervised Attack | PSNR | **32.55** |
| | SSIM | **0.91** |

Table 7: FGSM attack: Quantitative metrics on all test slices of Ax-FLAIR.

| Metric | SMUG | Adversarial Training (AT) | Proposed Method + MoDL / SMUG / AT |
|:---:|:---:|:---:|:---:|
| PSNR | **36.24** | 35.61 | **36.24** / 35.13 / 36.06 |
| SSIM | **0.93** | 0.93 | **0.93** / 0.92 / **0.93** |

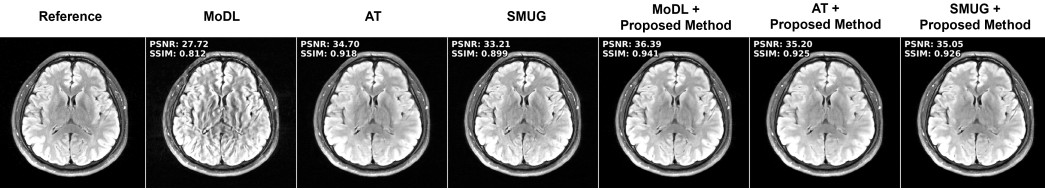

| Reference | MoDL | AT | SMUG | MoDL + Proposed Method | AT + Proposed Method | SMUG + Proposed Method |
|:---:|:---:|:---:|:---:|:---:|:---:|:---:|
| | PSNR: 27.72 SSIM: 0.812 | PSNR: 34.70 SSIM: 0.918 | PSNR: 33.21 SSIM: 0.899 | PSNR: 36.39 SSIM: 0.941 | PSNR: 35.20 SSIM: 0.925 | PSNR: 35.05 SSIM: 0.926 |

Figure 9: Performance of different methods under FGSM attack.

constructions with $\ell_2$ attacks in k-space using baseline MoDL, adversarial training and our proposed mitigation. Tab. 8 shows comparison of adversarial training and the proposed method on Cor-PD datasets, highlighting the efficacy of our method in this setup as well. We also emphasize that the $\ell_\infty$ image domain attacks are easily converted to attacks in k-space, which are non-zero only on indices specified by $\Omega$, as described in Section 2.2.

### D.8 Non-Uniform Undersampling Patterns

While the main text focused on uniform undersampling, which is considered to be a harder problem (Hammernik et al., 2018; Yaman et al., 2020), here we describe results with random undersampling, generated with a variable density Gaussian pattern (Aggarwal et al., 2019).

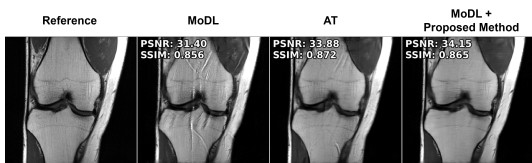

Figure 10: Representative reconstructions under $\ell_2$ attack on measurements with $\epsilon = 0.05 \cdot \|\mathbf{y}_\Omega\|_2$ using MoDL, adversarial training, and our proposed method.

All networks were retrained for such undersampling patterns. The attack generation and our mitigation algorithms were applied without any changes, as described in the main text. Fig. 11 shows representative examples for different methods, highlighting that our method readily extends to non-uniform undersampling patterns. Tab. 9 summarizes the quantitative metrics for this case, showing that the proposed mitigation improves upon MoDL or adversarial training alone.

## E  Blind Mitigation

This section shows that in addition to not needing any retraining for mitigation, our approach does not require precise information about how the attack is generated. Fig. 12 shows how the reconstruction improves as we use linear schedulers to find the optimum $(\epsilon, \alpha)$ values. Top row shows the tuning of $\epsilon$ while we keep

Table 8: Mitigation results for $\ell_2$ attacks in k-space.

| Method | Metric | $\ell_2$ Attack |
|---|---|---|
| Adversarial Training | PSNR | 33.37 |
| | SSIM | 0.88 |
| Proposed Method + MoDL | PSNR | **34.21** |
| | SSIM | **0.89** |

the step size $\alpha$ constant. After the cyclic loss in Eq. (10) stops decreasing, we fix this $\tilde{\epsilon}$ for the projection ball. The bottom row shows the effect of decreasing $\alpha$ for this $\tilde{\epsilon}$ value, from right to left. For this purpose, our linear scheduler for $\epsilon$ starts from 0.04 and decreases by 0.01 each step until the cyclic loss stabilizes. Then, step size $\alpha$ starts from a large value of $\epsilon$ and gradually decreases, ending at $\epsilon/3.5$ until the cyclic loss shows no further improvement.

As mentioned in the main text, since the $\ell_\infty$ ball contains the $\ell_2$ ball of the same radius, and noting the unitary nature of the Fourier transform in regards to $\ell_2$ attack strengths in k-space versus image domain, we always use the $\ell_\infty$ ball for blind mitigation. Furthermore, we provide results for using blind mitigation with $\ell_2$ attacks in k-space.

Table 9: Attacks on non-uniform undersampling.

| Metric | MoDL | Adversarial Training (AT) | Proposed Method + MoDL / AT |
|---|---|---|---|
| PSNR | 22.30 | 32.22 | 31.82 / **34.12** |
| SSIM | 0.62 | 0.89 | 0.87 / **0.92** |

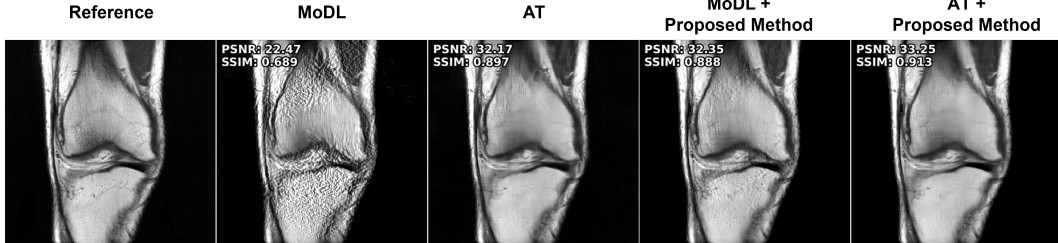

Figure 11: Representative reconstructions for non-uniform undersampling reconstructions using MoDL, adversarial training, and our proposed method under adversarial attacks.

Fig. 13 depicts example reconstructions with $\ell_2$ attacks in k-space using baseline MoDL and our blind mitigation approach. Tab. 10 compares our blind mitigation approach to our mitigation strategy with known attack type and level, showing that blind mitigation performs on-par with the latter for both $\ell_2$ attacks in k-space and $\ell_\infty$ attacks in image domain.

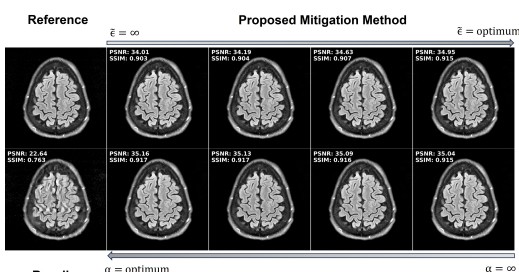

## F FURTHER DETAILS ON ADAPTIVE ATTACKS

This section contains more information about adaptive attack generation and visual examples. As discued earlier, strong performance against iterative optimization-based attacks is not nec-

Figure 12: Blind mitigation process of finding the optimum $(\epsilon, \alpha)$ parameters and corresponding results. Top row shows $\epsilon$ optimization for a fixed $\alpha$, while the bottom row shows $\alpha$ optimization for the optimum $\epsilon$. This joint optimization leads to a 1.15dB gain over the initial estimate.

essarily a good indicator of robustness and must be evaluated under adaptive attacks (Qiu et al., 2020; Guo et al., 2017; Prakash et al., 2018; Xie et al., 2017; Buckman et al., 2018). As mentioned in the main text, to generate the adaptive attack we unroll Algorithm 1 for $T$ iterations. The memory requirements of larger $T$ was handled by checkpointing (Chen et al., 2023; Kassis et al., 2024). Furthermore, the presence of $\tilde{\mathbf{n}}$ in Eq. (10) may suggest stochasticity in the system (Kassis et al., 2024). However, $\tilde{\mathbf{n}}$ is pre-calculated for a given input in our mitigation algorithm, and held constant throughout the mitigation. To make the adaptive attack as strong as possible, we pass this information about $\tilde{\mathbf{n}}$ to the adaptive attack as well, thus letting it have oracle knowledge about it. Finally, for maximal performance of the attack, we first tuned $\lambda$ in Eq. (11) empirically, then generated the adaptive attacks for $T \in \{10, 25, 50, 100\}$. Details on tuning of $\lambda$ and verification of gradient obfuscation avoidance in our adaptive attacks are detailed below.

### F.1 HYPERPARAMETER TUNING FOR ADAPTIVE ATTACKS

The parameter $\lambda$ in eq. (11) balances the two terms involved in the adaptive attack generation. A higher $\lambda$ produces a perturbation with more focus on bypassing the defense strategy, while potentially not generating a strong enough attack for the baseline. Conversely, a small $\lambda$ may not lead to sufficient adaptivity in the attack generation. To this end, we computed the population-average PSNRs of the reconstruction after the iterative mitigation algorithm on a subset of Cor-PD for various $\lambda$ values for $T \in \{10, 25\}$, as shown in Tab. 11.

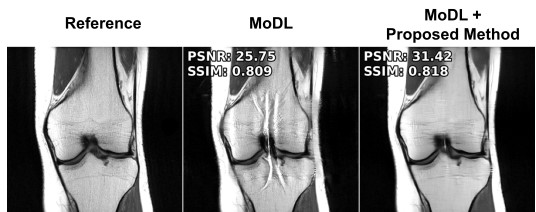

Figure 13: Representative reconstructions under $\ell_2$ attack using MoDL and our proposed blind mitigation.

These results show that $\lambda = 5$ leads to the most destructive attack against our mitigation algorithm, and was subsequently used for adaptive attack generation in Section 4.2.

### F.2 VERIFICATION OF GRADIENT OBFUSCATION AVOIDANCE

While our adaptive attack implements the exact gradient to avoid gradient obfuscation (including shattered, stochastic, and vanishing gradients (Athalye et al., 2018)), there are some methods to verify that gradients are indeed not obfuscated (Athalye et al., 2018). In par-

Table 10: Blind mitigation for $\ell_2$ (k-space, $\epsilon = 0.05 \cdot \|\mathbf{y}_\Omega\|_2$) and $\ell_\infty$ (image domain, $\epsilon = 0.01$) attacks on Cor-PD.

| Attack Method | Metric | Proposed Method ($\ell_\infty$ attack) | Proposed Method ($\ell_2$ attack) |
|---|---|---|---|
| Knowing the Attack | PSNR | **35.14** | 34.21 |
| | SSIM | **0.92** | **0.89** |
| Blind Mitigation | PSNR | 34.72 | 33.73 |
| | SSIM | **0.92** | 0.88 |

ticular, we tested two well-established key criteria: 1) One-step attacks should not outperform iterative-based ones, and 2) Increasing the perturbation bound (i.e. $\epsilon$) should lead to a greater disruption. Tab. 12 summarizes these two criteria, showing PSNRs of the iterative mitigation algorithm

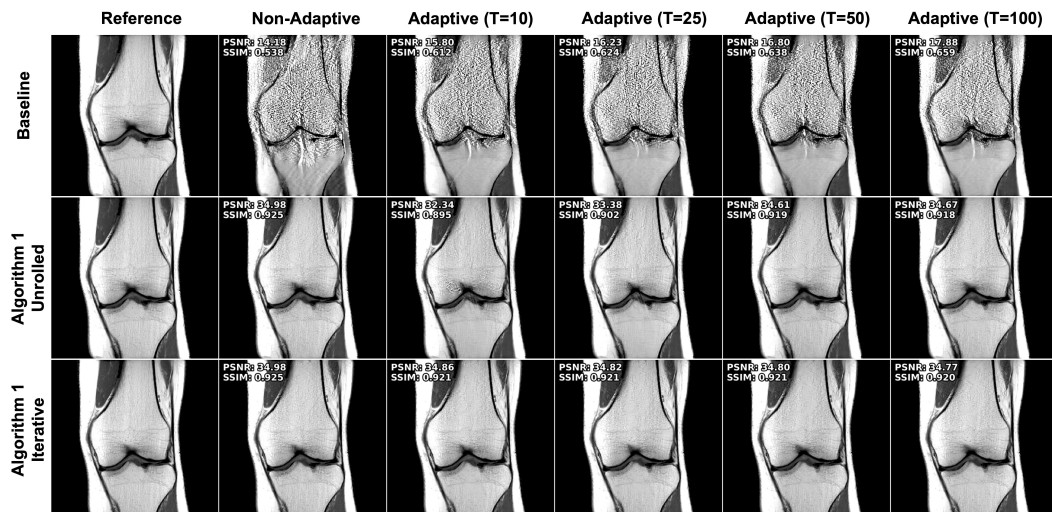

Figure 14: Representative examples of the mitigation algorithm outputs for adaptive attacks. The number of unrolls $T \in \{10, 25, 50, 100\}$ specified for each adaptive attack on the top. The top row is the baseline reconstruction, where the non-adaptive attack shows more artifacts than adaptive ones, as expected. The second row shows the mitigation outputs using the unrolled version of Algorithm 1, where the number of unrolls are matched between the adaptive attack generation and mitigation. At smaller $T$ values, the unrolled mitigation suffers from performance degradation. Finally, the last row shows the results of the iterative mitigation algorithm on the adaptive attacks. Iterative mitigation, when run until convergence, resolves the attacks, albeit with a slight degradation for high $T$ values. This is consistent with its physics-based design, showing its robustness to adaptive attacks.

output. These demonstrate that a single-step attack cannot surpass the iterative-based ones in terms of attack success, and similarly, increasing the perturbation bound leads to more severe degradation with PGD. These sanity checks align with the fact that we used the exact gradient through the steps described in Section 4.1, validating that gradient obfuscation did not happen in our implementation.

### F.3 VISUALIZATION OF ADAPTIVE ATTACKS AND MITIGATION

Representative examples showing the performance of the mitigation algorithm for different adaptive attacks generated using Eq. (11) with the unrolled version of $g(\cdot)$ for $T \in \{10, 25, 50, 100\}$ are provided in Fig. 14. The first row shows the results of the baseline reconstruction under both non-adaptive and adaptive attacks for various $T$. Consistent with Tab. 2, as $T$ increases for the adaptive attack, the baseline deterioration becomes less substantial. The second row shows the performance of the mitigation algorithm when it is unrolled for the same number of $T$ as in the adaptive attack genera-

Table 11: Fine-tuning the $\lambda$ parameter in Eq. (11) across $T \in \{10, 25\}$.

| Unrolls | $\lambda = 1$ | $\lambda = 2$ | $\lambda = 3$ | $\lambda = 5$ | $\lambda = 10$ |
|---|---|---|---|---|---|
| $T = 10$ | 34.51 | 34.48 | 34.41 | **34.34** | 34.66 |
| $T = 25$ | 34.41 | 34.36 | 34.40 | **34.16** | 34.47 |

Table 12: Checking gradient obfuscation on the Cor-PD dataset over $T \in \{10, 25\}$.

| $T$ | PGD ($\epsilon = 0.01$) | PGD ($\epsilon = 0.02$) | FGSM ($\epsilon = 0.01$) |
|---|---|---|---|
| 10 | 34.34 | 33.11 | 35.17 |
| 25 | 34.16 | 32.77 | 35.01 |

tion. In this case, for lower $T$, the unrolled mitigation has performance degradation, as expected. Finally, the final row shows results of the iterative mitigation algorithm run until convergence. In all cases, the iterative mitigation algorithm successfully recovers a clean image, owing to its physics-based nature, as discussed in Section 4.2. However, we note that though adaptive attacks have milder effect on the baseline with increasing $T$, they do deteriorate the iterative mitigation albeit slightly as a function of increasing $T$.

## G    PROOF OF THEOREM 1

The standard proof techniques for unrolled networks (Liang et al., 2023; Pramanik et al., 2023) performs an error propagation analysis through the layers of the network in image domain. However, this is insufficient in our case, as it does not capture differences between the behavior of the k-space in $\Omega$ and $\Omega^C$. Thus, here we present a different analysis approach.

We first present some basic notation and assumptions about multi-coil forward operator, $\mathbf{E}_\Omega \in \mathbb{C}^{M \times N}$ (Pruessmann et al., 1999; Lustig & Pauly, 2010; Uecker et al., 2014):

$$\mathbf{E}_\Omega = \begin{bmatrix} \mathbf{F}_\Omega \mathbf{S}_1 \\ \mathbf{F}_\Omega \mathbf{S}_2 \\ \vdots \\ \mathbf{F}_\Omega \mathbf{S}_{n_c} \end{bmatrix}, \tag{15}$$

where $\mathbf{F}_\Omega$ is a partial Fourier operator[1], $\mathbf{S}_k \triangleq \mathrm{diag}(\mathbf{s}_k)$ are diagonal matrices corresponding to the sensitivity map of the $k^{\text{th}}$ coil and $n_c$ is the number of coils. Note that, by design, coil sensitivity maps satisfy $\sum_k \mathbf{S}_k \mathbf{S}_k^H = \mathbf{I}$ (Uecker et al., 2014; Demirel et al., 2023a). We also define $\mathbf{E}_{\Omega^C} \in \mathbb{C}^{N \cdot n_c - M \times N}$ in an analogous manner. Finally, we note that $\mathbf{S}_k$ are smooth/low-frequency due to the physics of the MR acquisition (Pruessmann et al., 1999; Uecker et al., 2014), which will be defined more concretely below.

**Lemma 1.1.** *Let $\mathbf{S}_k = diag(\mathbf{s}_k)$ be smooth, defined as containing most of their energy in $L$ low-frequency coefficients in the Fourier domain, i.e. $\sum_{l \notin [-L/2, L/2-1]} |(Fs_k)_l|^2 < \zeta$, and assume $\Omega$ contains the ACS region with frequencies $[-L, L-1]$ Then*

$$||\mathbf{E}_\Omega \mathbf{E}_{\Omega^C}^H||_2 < c_1 \sqrt{\zeta} + c_2 \zeta$$

*fo constants $c_1 = 2n_c, c_2 = n_c/\sqrt{N}$.*

*Proof.* First note that in the single coil case, where $n_c = 1$ and $\mathbf{S}_1 = \mathbf{I}$, i.e. $\mathbf{E}_\Omega = \mathbf{F}_\Omega$ and $\mathbf{E}_{\Omega^C} = \mathbf{F}_{\Omega^C}$, we trivially have $\mathbf{F}_\Omega \mathbf{F}_{\Omega^C}^H = \mathbf{0}$ by the orthonormality of the rows of the discrete Fourier matrix. We will build on this intuition using the smoothness of $\mathbf{S}_k$ along the way. To this end, note

$$\mathbf{E}_\Omega \mathbf{E}_{\Omega^C}^H = \begin{bmatrix} \mathbf{F}_\Omega \mathbf{S}_1 \\ \mathbf{F}_\Omega \mathbf{S}_2 \\ \vdots \\ \mathbf{F}_\Omega \mathbf{S}_{n_c} \end{bmatrix} \begin{bmatrix} \mathbf{S}_1^H \mathbf{F}_{\Omega^C}^H & \mathbf{S}_2^H \mathbf{F}_{\Omega^C}^H & \dots & \mathbf{S}_{n_c}^H \mathbf{F}_{\Omega^C}^H \end{bmatrix} \tag{16}$$

has a block structure with $(p, q)^{\text{th}}$ block $\mathbf{B}_{pq}$ given by

$$\mathbf{B}_{pq} = \mathbf{F}_\Omega \mathbf{S}_p \mathbf{S}_q^H \mathbf{F}_{\Omega^C}^H.$$

Note by block Frobenius inequality

$$||\mathbf{E}_\Omega \mathbf{E}_{\Omega^C}^H||_2^2 \le \sum_{p,q} ||\mathbf{B}_{pq}||_2^2 \tag{17}$$

Next, we will consider the norm of the $(p, q)^{\text{th}}$ block:

$$\mathbf{F}_\Omega \mathbf{S}_p \mathbf{S}_q^H \mathbf{F}_{\Omega^C}^H = \mathbf{P}_\Omega \underbrace{\left( \mathbf{F} \mathbf{S}_p \mathbf{S}_q^H \mathbf{F}^H \right)}_{\triangleq \mathbf{C}_{pq}} \mathbf{P}_{\Omega^C}^H,$$

where $\mathbf{C}_{pq}$ implements a circulant matrix implementing a circular convolution operation with kernel

$$\mathbf{b}_{pq} = \mathbf{F}(\mathbf{s}_p \odot \bar{\mathbf{s}}_q) = \frac{1}{\sqrt{N}} \underbrace{\mathbf{F}\mathbf{s}_p}_{\mathbf{a}_p} \circledast \underbrace{\mathbf{F}\bar{\mathbf{s}}_q}_{\bar{\mathbf{a}}_q}.$$

---

[1]We will use $\mathbf{F} \in \mathbb{C}^{N \times N}$ to denote the full discrete Fourier transform matrix. This allows us to equivalently define $\mathbf{F}_\Omega \triangleq \mathbf{P}_\Omega \mathbf{F}$, where $\mathbf{P}_\Omega$ is a binary mask specifying the sampling pattern $\Omega$.

Here $\bar{\cdot}$ denotes elementwise complex conjugation, $\odot$ denotes the elementwise Hadamard product and $\circledast$ denotes circular convolution. Now note

$$||\mathbf{B}_{pq}||_2 \leq ||\mathbf{C}_{pq}||_2 = \max_j |(\mathbf{b}_{pq})_j| = \max_j \left| \frac{1}{\sqrt{N}} \sum_l \mathbf{a}_{p,l} \overline{\mathbf{a}_{q,l-j}} \right|$$

To calculate this last term, we decompose $\mathbf{a}_p$ and $\mathbf{a}_q$ into their low-frequency (between $[-L/2, L/2 - 1]$) and the remaining high-frequency components, noting $||\mathbf{a}_k^{\text{high}}||_2^2 \leq \zeta$. Then

$$(\mathbf{b}_{pq})_j = \frac{1}{\sqrt{N}} \left[ (\mathbf{a}_p^{\text{low}} \circledast \overline{\mathbf{a}_q^{\text{low}}})_j + (\mathbf{a}_p^{\text{low}} \circledast \overline{\mathbf{a}_q^{\text{high}}})_j + (\mathbf{a}_p^{\text{high}} \circledast \overline{\mathbf{a}_q^{\text{low}}})_j + (\mathbf{a}_p^{\text{high}} \circledast \overline{\mathbf{a}_q^{\text{high}}})_j \right].$$

Using Cauchy-Schwarz inequality:

$$|(\mathbf{a}_p^{\text{low}} \circledast \overline{\mathbf{a}_q^{\text{high}}})_j| \leq ||\mathbf{a}_p^{\text{low}}||_2 ||\mathbf{a}_q^{\text{high}}||_2 < \sqrt{\zeta N}.$$

Similarly for the two high-frequency terms:

$$|(\mathbf{a}_p^{\text{high}} \circledast \overline{\mathbf{a}_q^{\text{high}}})_j| \leq ||\mathbf{a}_p^{\text{high}}||_2 ||\mathbf{a}_q^{\text{high}}||_2 < \zeta.$$

Finally the convolution of the two low-frequency terms are supported between $[-L, L-1]$. Since all these frequencies are in $\Omega$, then these vanish for the values picked up in $\mathbf{B}_{pq}$ by the corresponding masks, leading to

$$||\mathbf{B}_{pq}||_2 < 2\sqrt{\zeta} + \zeta/\sqrt{N}.$$

Combining this with Eq. (17) yields

$$||\mathbf{E}_\Omega \mathbf{E}_{\Omega^C}^H||_2 < n_c(2\sqrt{\zeta} + \zeta/\sqrt{N}). \tag{18}$$

$\square$

**Corollary 1.1.** *Under the same conditions, we have* $||E_{\Omega^c}||_2 < n_c \zeta/\sqrt{N}$.

This corollary is helpful in establishing

$$||\mathbf{E}_\Omega \mathbf{x}||_2^2 = ||\mathbf{x}||_2^2 - ||\mathbf{E}_{\Omega^C} \mathbf{x}||_2^2 \Longrightarrow ||\mathbf{x}||_2 \leq \frac{||\mathbf{E}_\Omega \mathbf{x}||_2}{\sqrt{1 - ||\mathbf{E}_{\Omega^C}||_2^2}}, \tag{19}$$

for $n_c \zeta < \sqrt{N}$. Note trivially $||E_{\Omega^c}||_2 \leq 1$ by the properties of the discrete Fourier transform and since $\sum_k \mathbf{S}_K \mathbf{S}_K^H = \mathbf{I}$. We also note in our experiments $n_c \leq 20$, while $\sqrt{N} \geq 320$.

**Lemma 1.2.** $\mathbf{E}_\Omega(\mathbf{E}_\Omega^H \mathbf{E}_\Omega + \mu\mathbf{I})^{-1} = (\mu\mathbf{I} + \mathbf{E}_\Omega \mathbf{E}_\Omega^H)^{-1} \mathbf{E}_\Omega$

*Proof.* By Woodbury's matrix identity, we have

$$(\mathbf{E}_\Omega^H \mathbf{E}_\Omega + \mu\mathbf{I})^{-1} = \frac{\mathbf{I}}{\mu} - \frac{\mathbf{I}}{\mu} \mathbf{E}_\Omega^H (\mu\mathbf{I} + \mathbf{E}_\Omega \mathbf{E}_\Omega^H)^{-1} \mathbf{E}_\Omega \tag{20}$$

Then

$$\mathbf{E}_\Omega(\mathbf{E}_\Omega^H \mathbf{E}_\Omega + \mu\mathbf{I})^{-1} = \frac{\mathbf{E}_\Omega}{\mu} - \frac{\mathbf{E}_\Omega \mathbf{E}_\Omega^H}{\mu}(\mu\mathbf{I} + \mathbf{E}_\Omega \mathbf{E}_\Omega^H)^{-1} \mathbf{E}_\Omega$$

$$= \left( \frac{\mu\mathbf{I} + \mathbf{E}_\Omega \mathbf{E}_\Omega^H}{\mu} - \frac{\mathbf{E}_\Omega \mathbf{E}_\Omega^H}{\mu} \right)(\mu\mathbf{I} + \mathbf{E}_\Omega \mathbf{E}_\Omega^H)^{-1} \mathbf{E}_\Omega$$

$$= (\mu\mathbf{I} + \mathbf{E}_\Omega \mathbf{E}_\Omega^H)^{-1} \mathbf{E}_\Omega$$

$\square$

**Proof of Theorem 1.** We will next do our error propagation analysis. We will use $\mathbf{x} = \mathbf{E}_\Omega^H \mathbf{E}_\Omega \mathbf{x} + \mathbf{E}_{\Omega^C}^H \mathbf{E}_{\Omega^C} \mathbf{x}$ to decompose the contributions from $\Omega$ and $\Omega^C$ frequencies. Note the former are always brought back close to the measurements due to the presence of DF units, while the latter does not follow this behavior. Let $\mathbf{x}^{(k)}$ and $\mathbf{z}^{(k)}$ denote the DF unit output and the proximal operator output

of the $k^{\text{th}}$ unroll of the PD-DL network for the *clean input* $\mathbf{y}_\Omega$. Similarly $\tilde{\mathbf{x}}^{(k)}$ and $\tilde{\mathbf{z}}^{(k)}$ denote the DF unit output and the proximal operator output of the $k^{\text{th}}$ unroll of the PD-DL network for the *perturbed input* $\mathbf{y}_\Omega + \mathbf{w}$. Analogously, we will define $\mathbf{y}_\Omega^{(k)} = \mathbf{E}_\Omega \mathbf{x}^{(k)}$ and $\tilde{\mathbf{y}}_\Omega^{(k)} = \mathbf{E}_\Omega \tilde{\mathbf{x}}^{(k)}$. We will also use singular value decompositions for $\mathbf{E}_\Omega = \mathbf{U}\boldsymbol{\Sigma}_\Omega \mathbf{V}^H$ and $\mathbf{E}_{\Omega^C} = \mathbf{U}'\boldsymbol{\Sigma}_{\Omega^C}\mathbf{V}'^H$. Finally, we will denote the largest and smallest singular values of $\mathbf{E}_\Omega$ by $\sigma_{\max}^\Omega$ and $\sigma_{\min}^\Omega$ respectively, and those of $\mathbf{E}_{\Omega^C}$ by $\sigma_{\max}^{\Omega^C}$ and $\sigma_{\min}^{\Omega^C}$ analogously. With these in place, we have:

$$
\begin{aligned}
\left\|(\tilde{\mathbf{y}}_\Omega^{(k)} - \mathbf{y}_\Omega^{(k)})\right\|_2 &= \left\|\mathbf{E}_\Omega(\tilde{\mathbf{x}}^{(k)} - \mathbf{x}^{(k)})\right\|_2 \\
&= \left\|\mathbf{E}_\Omega\left(\mathbf{E}_\Omega^H \mathbf{E}_\Omega + \mu \mathbf{I}\right)^{-1}\left[(\mathbf{E}_\Omega^H \tilde{\mathbf{y}}_\Omega + \mu \tilde{\mathbf{z}}^{(k)}) - (\mathbf{E}_\Omega^H \mathbf{y}_\Omega + \mu \mathbf{z}^{(k)})\right]\right\|_2 \\
&= \left\|\mathbf{E}_\Omega\left(\mathbf{E}_\Omega^H \mathbf{E}_\Omega + \mu \mathbf{I}\right)^{-1}\left[\mathbf{E}_\Omega^H \mathbf{w} + \mu(\tilde{\mathbf{z}}^{(k)} - \mathbf{z}^{(k)})\right]\right\|_2 \\
&= \left\|\mathbf{E}_\Omega\left(\mathbf{E}_\Omega^H \mathbf{E}_\Omega + \mu \mathbf{I}\right)^{-1}\left[\mathbf{E}_\Omega^H \mathbf{w} + \mu \mathbf{E}_\Omega^H \mathbf{E}_\Omega(\tilde{\mathbf{z}}^{(k)} - \mathbf{z}^{(k)})\right] + \right. \\
&\qquad \left. \mathbf{E}_\Omega\left(\mathbf{E}_\Omega^H \mathbf{E}_\Omega + \mu \mathbf{I}\right)^{-1}\left[\mu \mathbf{E}_{\Omega^C}^H \mathbf{E}_{\Omega^C}(\tilde{\mathbf{z}}^{(k)} - \mathbf{z}^{(k)})\right]\right\|_2 \\
&\leq \left\|\mathbf{E}_\Omega\left(\mathbf{E}_\Omega^H \mathbf{E}_\Omega + \mu \mathbf{I}\right)^{-1}\left[\mathbf{E}_\Omega^H \mathbf{w} + \mu \mathbf{E}_\Omega^H \mathbf{E}_\Omega(\tilde{\mathbf{z}}^{(k)} - \mathbf{z}^{(k)})\right]\right\|_2 + \\
&\qquad \left\|\mathbf{E}_\Omega\left(\mathbf{E}_\Omega^H \mathbf{E}_\Omega + \mu \mathbf{I}\right)^{-1}\left[\mu \mathbf{E}_{\Omega^C}^H \mathbf{E}_{\Omega^C}(\tilde{\mathbf{z}}^{(k)} - \mathbf{z}^{(k)})\right]\right\|_2.
\end{aligned} \tag{21}
$$

Now we derive a bound for the second term, which characterizes the effect of $\mathbf{E}_\Omega$ on the contributions from $\mathbf{E}_{\Omega^C}^H \mathbf{E}_{\Omega^C}$:

$$
\begin{aligned}
&\left\|\mathbf{E}_\Omega\left(\mathbf{E}_\Omega^H \mathbf{E}_\Omega + \mu \mathbf{I}\right)^{-1}\left[\mu \mathbf{E}_{\Omega^C}^H \mathbf{E}_{\Omega^C}(\tilde{\mathbf{z}}^{(k)} - \mathbf{z}^{(k)})\right]\right\|_2 \\
\xrightarrow{\text{Lemma 1.2}} &= \left\|(\mu \mathbf{I} + \mathbf{E}_\Omega \mathbf{E}_\Omega^H)^{-1}\mathbf{E}_\Omega\left[\mu \mathbf{E}_{\Omega^C}^H \mathbf{E}_{\Omega^C}(\tilde{\mathbf{z}}^{(k)} - \mathbf{z}^{(k)})\right]\right\|_2 \\
&\leq \mu \left\|(\mu \mathbf{I} + \mathbf{E}_\Omega \mathbf{E}_\Omega^H)^{-1}\right\|_2 \left\|\mathbf{E}_\Omega \mathbf{E}_{\Omega^C}^H\right\|_2 \left\|\mathbf{E}_{\Omega^C}(\tilde{\mathbf{z}}^{(k)} - \mathbf{z}^{(k)})\right\|_2 \\
&\leq \mu \left\|(\mu \mathbf{I} + \mathbf{U}\boldsymbol{\Sigma}_\Omega \mathbf{V}^H \mathbf{V}\boldsymbol{\Sigma}_\Omega \mathbf{U}^H)^{-1}\right\|_2 \left\|\mathbf{E}_\Omega \mathbf{E}_{\Omega^C}^H\right\|_2 \left\|\mathbf{E}_{\Omega^C}\right\|_2 m \left\|\tilde{\mathbf{x}}^{(k-1)} - \mathbf{x}^{(k-1)}\right\|_2 \\
&\leq \mu \left\|\mathbf{U}(\boldsymbol{\Sigma}_\Omega^2 + \mu \mathbf{I})^{-1}\mathbf{U}^H\right\|_2 \left\|\mathbf{E}_\Omega \mathbf{E}_{\Omega^C}^H\right\|_2 \left\|\mathbf{E}_{\Omega^C}\right\|_2 m \|\tilde{\mathbf{x}}^{(k-1)} - \mathbf{x}^{(k-1)}\|_2 \\
\xrightarrow{\text{Corollary 1.1}} &\leq \mu \left\|(\boldsymbol{\Sigma}_\Omega^2 + \mu \mathbf{I})^{-1}\right\|_2 \left\|\mathbf{E}_\Omega \mathbf{E}_{\Omega^C}^H\right\|_2 \frac{m\left\|\mathbf{E}_{\Omega^C}\right\|_2}{\sqrt{1 - \|\mathbf{E}_{\Omega^C}\|_2^2}} \|\tilde{\mathbf{y}}_\Omega^{(k-1)} - \mathbf{y}_\Omega^{(k-1)}\|_2 \\
&\leq \frac{\mu}{\mu + (\sigma_{\min}^\Omega)^2} \left\|\mathbf{E}_\Omega \mathbf{E}_{\Omega^C}^H\right\|_2 \frac{m\sigma_{\max}^{\Omega^C}}{\sqrt{1 - \sigma_{\max}^{\Omega^C}}} \|\tilde{\mathbf{y}}_\Omega^{(k-1)} - \mathbf{y}_\Omega^{(k-1)}\|_2 \\
\xrightarrow{\text{Lemma 1.1}} &\leq \underbrace{\frac{\mu}{\mu + (\sigma_{min}^\Omega)^2}\left(2n_c\sqrt{\zeta} + \frac{\zeta}{\sqrt{N}}\right)\frac{m\sigma_{\max}^{\Omega^C}}{\sqrt{1 - \sigma_{\max}^{\Omega^C}}}}_{\alpha_\Omega} \|\tilde{\mathbf{y}}_\Omega^{(k-1)} - \mathbf{y}_\Omega^{(k-1)}\|_2
\end{aligned} \tag{22}
$$

Next we consider the first term of Eq. (21), which characterizes the effect of $\mathbf{E}_\Omega$ on the contributions from $\mathbf{E}_\Omega^H \mathbf{E}_\Omega$:

$$
\begin{aligned}
&\left\|\mathbf{E}_\Omega\left(\mathbf{E}_\Omega^H \mathbf{E}_\Omega + \mu \mathbf{I}\right)^{-1}\left[\mathbf{E}_\Omega^H \mathbf{w} + \mu \mathbf{E}_\Omega^H \mathbf{E}_\Omega(\tilde{\mathbf{z}}^{(k)} - \mathbf{z}^{(k)})\right]\right\|_2 \\
&\leq \left\|\mathbf{E}_\Omega\left(\mathbf{E}_\Omega^H \mathbf{E}_\Omega + \mu \mathbf{I}\right)^{-1}\mathbf{E}_\Omega^H \mathbf{w}\right\|_2 + \mu \left\|\mathbf{E}_\Omega\left(\mathbf{E}_\Omega^H \mathbf{E}_\Omega + \mu \mathbf{I}\right)^{-1}\mathbf{E}_\Omega^H \mathbf{E}_\Omega(\tilde{\mathbf{z}}^{(k)} - \mathbf{z}^{(k)})\right\|_2 \\
&\leq \left\|\mathbf{E}_\Omega\left(\mathbf{E}_\Omega^H \mathbf{E}_\Omega + \mu \mathbf{I}\right)^{-1}\mathbf{E}_\Omega^H \mathbf{w}\right\|_2 + \mu \left\|\mathbf{E}_\Omega\left(\mathbf{E}_\Omega^H \mathbf{E}_\Omega + \mu \mathbf{I}\right)^{-1}\mathbf{E}_\Omega^H \mathbf{E}_\Omega\right\|_2 \left\|(\tilde{\mathbf{z}}^{(k)} - \mathbf{z}^{(k)})\right\|_2 \\
&\leq \left\|\mathbf{E}_\Omega\left(\mathbf{E}_\Omega^H \mathbf{E}_\Omega + \mu \mathbf{I}\right)^{-1}\mathbf{E}_\Omega^H \mathbf{w}\right\|_2 + \mu \left\|\mathbf{E}_\Omega\left(\mathbf{E}_\Omega^H \mathbf{E}_\Omega + \mu \mathbf{I}\right)^{-1}\mathbf{E}_\Omega^H \mathbf{E}_\Omega\right\|_2 m \left\|(\tilde{\mathbf{x}}^{(k-1)} - \mathbf{x}^{(k-1)})\right\|_2
\end{aligned}
$$

$$\leq \left\| \mathbf{U}\boldsymbol{\Sigma}_\Omega \mathbf{V}^H \left( \mathbf{V}\boldsymbol{\Sigma}_\Omega \mathbf{U}^H \mathbf{U}\boldsymbol{\Sigma}_\Omega \mathbf{V}^H + \mu\mathbf{I} \right)^{-1} \mathbf{V}\boldsymbol{\Sigma}_\Omega \mathbf{U}^H \mathbf{w} \right\|_2$$

$$+ \left\| \mathbf{U}\boldsymbol{\Sigma}_\Omega \mathbf{V}^H \left( \mathbf{V}\boldsymbol{\Sigma}_\Omega \mathbf{U}^H \mathbf{U}\boldsymbol{\Sigma}_\Omega \mathbf{V}^H + \mu\mathbf{I} \right)^{-1} \mathbf{V}\boldsymbol{\Sigma}_\Omega \mathbf{U}^H \mathbf{U}\boldsymbol{\Sigma}_\Omega \mathbf{V}^H \right\|_2 m \left\| (\tilde{\mathbf{x}}^{(k-1)} - \mathbf{x}^{(k-1)}) \right\|_2$$

$$\leq \|\boldsymbol{\Sigma}_\Omega\|_2^2 \|(\boldsymbol{\Sigma}_\Omega^2 + \mu\mathbf{I})^{-1}\|_2 \|\mathbf{w}\|_2 + \frac{m\|\boldsymbol{\Sigma}_\Omega\|_2^3 \|(\boldsymbol{\Sigma}_\Omega^2 + \mu\mathbf{I})^{-1}\|_2}{\sqrt{1 - \|\mathbf{E}_{\Omega^C}\|_2^2}} \left\| \tilde{\mathbf{y}}_\Omega^{(k-1)} - \mathbf{y}_\Omega^{(k-1)} \right\|_2$$

$$\leq \underbrace{\frac{(\sigma_{max}^\Omega)^2}{(\sigma_{min}^\Omega)^2 + \mu}}_{\beta} \|\mathbf{w}\|_2 + \underbrace{\frac{m(\sigma_{max}^\Omega)^3}{(\sigma_{min}^\Omega)^2 + \mu} \cdot \sqrt{\frac{1}{1 - (\sigma_{max}^{\Omega^C})^2}}}_{\alpha_\Omega} \left\| \tilde{\mathbf{y}}_\Omega^{(k-1)} - \mathbf{y}_\Omega^{(k-1)} \right\|_2 \tag{23}$$

Combining these with Eq. (21), the recursive relation across unrolls is given by:

$$\left\| \tilde{\mathbf{y}}_\Omega^{(k)} - \mathbf{y}_\Omega^{(k)} \right\|_2 \leq \beta \|\mathbf{w}\|_2 + (\alpha_\Omega + \alpha_{\Omega^C}) \left\| \tilde{\mathbf{y}}_\Omega^{(k-1)} - \mathbf{y}_\Omega^{(k-1)} \right\|_2 \tag{24}$$

Evaluating the recursion through the $K$ unrolls yields:

$$\|\mathbf{E}_\Omega(\tilde{\mathbf{x}} - \mathbf{x})\|_2 = \left\| \tilde{\mathbf{y}}_\Omega^{(K)} - \mathbf{y}_\Omega^{(K)} \right\|_2 \leq \left( \beta \frac{1 - (\alpha_\Omega + \alpha_{\Omega^C})^K}{1 - (\alpha_\Omega + \alpha_{\Omega^C})} + (\alpha_\Omega + \alpha_{\Omega^C})^K \right) \|\mathbf{w}\|_2 \tag{25}$$

## H ABLATION STUDY

As discussed in Section 4.3, we analyzed the number of reconstruction stages for mitigation. By extending the number of reconstruction stages, we can reformulate this by updating the second term in the loss function in Eq. (10) to include more reconstruction stages, for instance with 3 cyclic stages instead of 2 given in Eq. (10):

$$\arg\min_{\mathbf{r}':\|\mathbf{r}'\|_p \leq \epsilon} \mathbb{E}_\Gamma \mathbb{E}_\Delta \Bigg[ \Bigg\| (\mathbf{E}_\Omega^H)^\dagger (\mathbf{z}_\Omega + \mathbf{r}') -$$

$$\mathbf{E}_\Omega f \Big( \mathbf{E}_\Gamma^H \Big( \mathbf{E}_\Gamma f \big( \mathbf{E}_\Delta^H \big( \mathbf{E}_\Delta f(\mathbf{z}_\Omega + \mathbf{r}', \mathbf{E}_\Omega; \boldsymbol{\theta}) + \tilde{\mathbf{n}} \big) \big), \mathbf{E}_\Delta; \boldsymbol{\theta} \Big) + \tilde{\mathbf{n}}, \mathbf{E}_\Gamma \Big); \boldsymbol{\theta} \Big) \Bigg\|_2 \Bigg]. \tag{26}$$

Empirically, in our implementation, we carry out the expectation over all possible permutations without repeating any patterns. As a result, the error propagated to the last stage becomes larger, as we rely more on synthesized data. In turn, this makes the optimization process harder, deteriorating the results, as shown in Fig. 15. Consequently, in addition to these performance issues, the computation costs of adding more cyclic reconstruction is often impractical, leading to the conclusion that 2-cyclic stages as in Eq. (10) are sufficient.

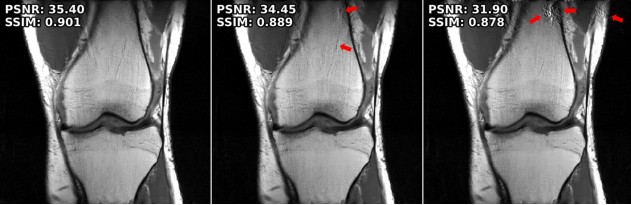

Figure 15: Ablation study on the number of stages for cyclic measurement consistency. Two reconstruction levels (left) outperform deeper variants (middle, right), as additional stages overly rely on synthesized k-space data.

# I COMPARISONS TO OTHER TRAINING-FREE DEFENSE METHODOLOGIES

This section explores other training-free mitigation approaches against adversarial attacks and their implementation details. We compare our method to JPEG compression, total-variation (TV) minimization, randomized smoothing (RS) (Cohen et al., 2019) on the input.

**JPEG Compression.** This is an input transform method that is popular for defending against adversarial attacks (Aydemir et al., 2018; Cucu et al., 2023). JPEG uses discrete cosine transform (DCT) for image compression, which inherently modifies the perturbed input without substantially changing its visual characteristics, potentially away from its worst-case behavior. We adopt the standard JPEG compression pipeline for complex-valued MRI images by applying the transform separately to the positive and negative parts of both the real and imaginary components of $\mathbf{z}_\Omega^{\mathrm{p}} = \mathbf{z}_\Omega + \mathbf{r}$. The four compressed channels are recombined to reconstruct the complex-valued data.

**TV Minimization/Denoising** This is another input–space transformation that has been employed against adversarial perturbations (Sheikh & Zafar, 2024). In particular, it optimizes the following denoising objective:

$$\min_{\mathbf{z}} \|\mathbf{z}_\Omega^{\mathrm{p}} - \mathbf{z}\|_2^2 + \beta TV(\mathbf{z}) \tag{27}$$

where the $TV(\cdot)$ is the total-variation norm formulated as follow:

$$TV(\mathbf{x}) = \sum_{i,j} \left( |\mathbf{x}_{i+1,j} - \mathbf{x}_{i,j}| + |\mathbf{x}_{i,j+1} - \mathbf{x}_{i,j}| \right). \tag{28}$$

We followed this formulation to denoise the adversarially perturbed input. In particular, given the perturbed input $\mathbf{z}_\Omega^{\mathrm{p}}$ we solve Eq. (27) using $\lambda = 0.01$ and 5 iterations.

Note that the application of TV denoising as a defense mechanism for MRI reconstruction is not straightforward, as the input image $\mathbf{z}_\Omega$ is inherently aliased due to undersampling. In contrast, TV regularization guides its output to a piecewise smooth output to minimize the objective in Eq. (27). Thus if run with a large $\lambda$ or for a large number of iterations, the aliasing artifacts will be changed, breaking the relationship with the forward operator $\mathbf{E}_\Omega$ in Eq. (1). Note a similar concern applies to other image-domain denoising-based input purification strategies, such as diffusion purification (Alkhouri et al., 2023; 2024) discussed further below, as the diffusion purification will push the output to the clean data manifold, losing the aliasing information that defines the relationship in Eq. (1). In summary, for TV minimization, the iteration count was intentionally kept small, as further optimization begins to de-alias the zero-filled image, further degrading the overall quality of the subsequent PD-DL reconstruction.

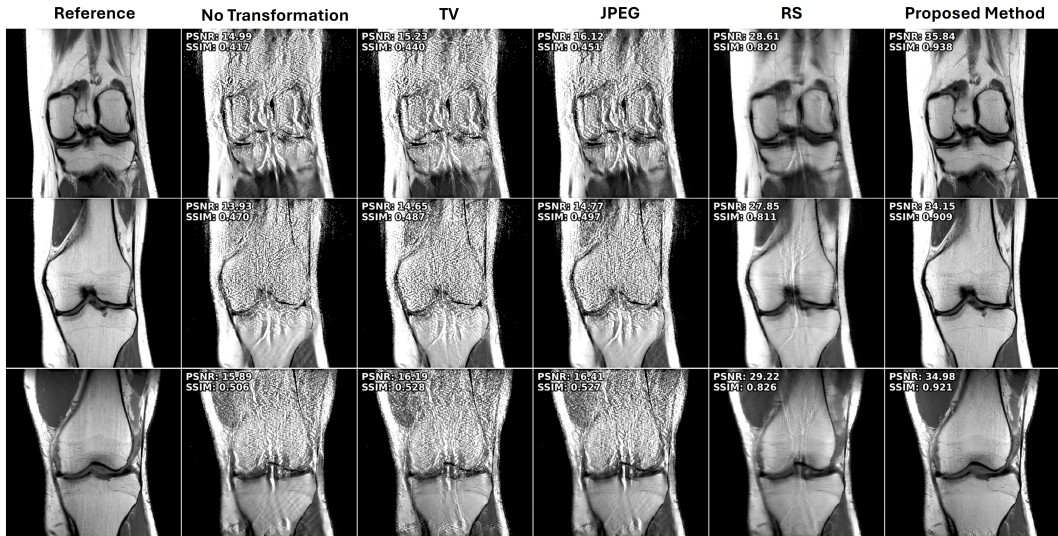

Figure 16: Representative results of different training-free methodologies applied on the perturbed inputs. Reconstructions show the proposed method outperforms existing classical approaches.

Table 13: $\lambda$ fine-tuning

| Metric | MoDL | $\lambda = 0.01$ | $\lambda = 0.02$ | $\lambda = 0.05$ | $\lambda = 0.1$ | $\lambda = 0.2$ | $\lambda = 0.3$ |
|--------|------|------------------|------------------|------------------|------------------|------------------|------------------|
| PSNR | 16.30 | 22.92 | 23.05 | 24.09 | 27.38 | **30.71** | 29.65 |
| SSIM | 0.486 | 0.622 | 0.628 | 0.670 | 0.777 | **0.858** | 0.851 |

**Randomized Smoothing (RS)** RS was first introduced for classification tasks (Cohen et al., 2019) and was later extended to regression settings (Rekavandi et al., 2024). In MRI reconstruction, given a pre-trained network $f(\cdot, \cdot; \boldsymbol{\theta})$, RS is performed as (Liang et al., 2023):

$$f_{\text{smooth}}(\mathbf{z}_\Omega^{\text{P}}, \mathbf{E}_\Omega; \boldsymbol{\theta}) = \mathbb{E}_{\boldsymbol{\xi} \sim \mathcal{N}(\mathbf{0}, \lambda\mathbf{I})}\left[f(\mathbf{E}_\Omega^{\text{H}}(\mathbf{y}_\Omega^{\text{P}} + \boldsymbol{\xi}), \mathbf{E}_\Omega; \boldsymbol{\theta})\right] \tag{29}$$

where $\lambda$ is a tuning parameter that control the trade-off between accuracy and robustness. In our implementation, we generated 50 zero-mean Gaussian noise samples, added them to the perturbed k-space representation $\mathbf{y}_\Omega^{\text{P}}$, and reconstructed each using the pre-trained MoDL. The final output was obtained by averaging these reconstructions. For noise injection, $\lambda$ was adjusted to fine-tune performance across the dataset. Tab. 13 shows the population metrics on different $\lambda$ values. Representative reconstruction results and the population metrics of each method on the test set are shown in Fig. 16 and Tab. 14, respectively. Both JPEG and TV denoising methods fail to mitigate the attack in a meaningful manner. RS outperforms these two input transformation methods, but still substantially falls short of the proposed method both quantitatively and visually with extensive blurring due to the high-level noise and inherent averaging.

## J COMPARISON TO DIFFUSION PURIFICATION FOR MRI RECONSTRUCTION

Diffusion purification (Nie et al., 2022) is a technique proposed to mitigate the effect of adversarial samples before they are used in downstream tasks. In particular, it works by adding scheduled noise to an adversarial example up to a diffusion step $t_1$, causing the clean and adversarial distributions to progressively converge (e.g. at $t_1 = T$, both distributions become standard Gaussian). Then the method performs $t_1$ reverse diffusion steps to purify the corrupted adversarial sample. In this way, purification enhances robustness in tasks such as classification (Nie et al., 2022). However, the application of diffusion purification for MRI reconstruction is not straightforward. As mentioned in the TV denoising subsection, diffusion models are trained to learn the distribution of clean fully-sampled data. Thus, if one noises the undersampled zerofilled images (aliased data) and tries to run the reverse diffusion process on this, it will inevitably push the output towards the clean data manifold, which will in turn no longer be consistent with the forward model (and the inverse problem objective). An early work (Alkhouri et al., 2024) considered diffusion purification for the MRI reconstruction problem studied here, and showed that even after purification, a fine-tuning step is required on MoDL network to handle the mismatch between the aliased images and score model trained on clean images. A more comprehensive version of this work (Alkhouri et al., 2023) by the same authors further replaced the noising-denoising purification with a full diffusion model-based ScoreMRI reconstruction (Chung & Ye, 2022), combined with yet another fine-tuned MoDL network. Note in both setups: 1) There is an inherent mismatch between attack generation (on baseline MoDL network) and reconstruction (with two separate reconstructions including ScoreMRI and fine-tuned MoDL), 2) The method is not training-free, as it requires fine-tuning of the MoDL network, 3) The method was never verified against adaptive attacks, which is the gold standard attack technique considered extensively in our work.

Table 14: Comparison of different input-transformation defenses on the Cor-PD dataset, in presence of perturbation. Mean of the population metrics are reported.

| Dataset | Metric | MoDL | Input TV + MoDL | Input JPEG + MoDL | Input RS + MoDL | Proposed Method + MoDL |
|---------|--------|------|-----------------|-------------------|-----------------|------------------------|
| Cor-PD | PSNR | 16.30 | 16.36 | 17.29 | 30.71 | **35.14** |
|         | SSIM | 0.486 | 0.488 | 0.509 | 0.858 | **0.921** |

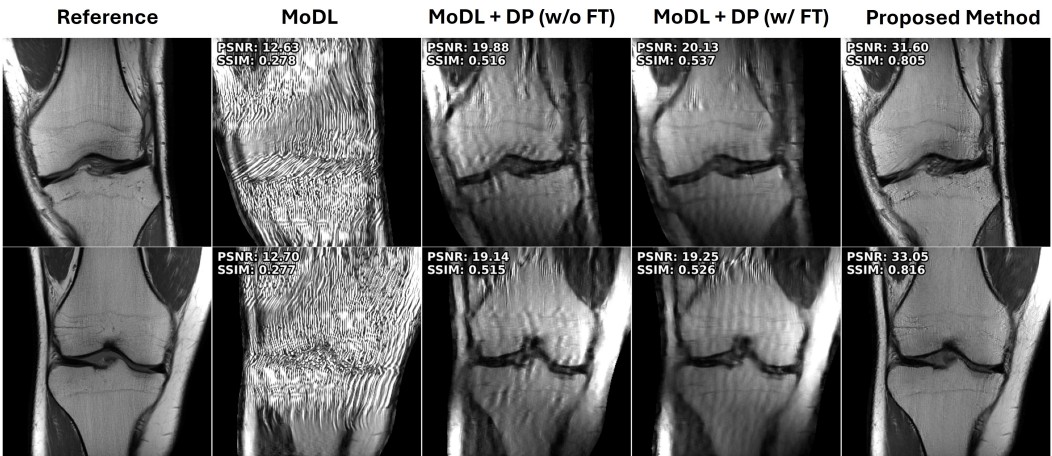

Figure 17: Comparison of proposed method with diffusion purification (Alkhouri et al., 2023). While fine-tuning improves the reconstruction quality of the diffusion purification approach, it still does not surpass the performance of our proposed method.

Nonetheless, even though these approaches (Alkhouri et al., 2023; 2024) are not training-free, to provide a more complete comparison, we additionally implemented diffusion-based adversarial purification pipelines in (Alkhouri et al., 2023), which as discussed earlier is the more comprehensive preprint extending on (Alkhouri et al., 2024). In particular, we reproduced the diffusion purification pipeline , using our own training sets and unrolled reconstruction models, while following the purification description in (Alkhouri et al., 2023). To do so, we first retrained MoDL, as our baseline $f(\cdot, \cdot; \boldsymbol{\theta})$, on Cor-PD knee data at a matrix size of $320 \times 320$ (instead of $332 \times 320$ used in the main text) to ensure compatibility with the matrix sizes expected by the pretrained diffusion model. An acceleration rate of 4 with a random sampling mask, with 24 ACS center lines was used. Second, we generated noisy samples by adding Gaussian noise to the clean zerofilled images, and then passed them through diffusion purification $DP_\phi(\cdot, \cdot, \cdot)$ step described in Alg.2 (Alkhouri et al., 2023) to obtain the purified outputs.

Here $\phi$ specifies the diffusion model (DM) parameters. In particular:

$$\mathbf{z}_\Omega^{\text{pur}} = \boldsymbol{DP}_\phi(\mathbf{z}_\Omega + \mathbf{w}, \mathbf{E}_\Omega, N_r)$$

where $\mathbf{w} \sim \mathcal{N}(\mathbf{0}, \sigma^2 \mathbf{I})$ with $\sigma = 0.001$, and $N_r = 150$ is the tuned parameter for corruption/purification. These purified sampled is then used to fine-tune the

Table 15: Comparison of MoDL, diffusion purification (with and without fine-tuning), and our proposed method on the Cor-PD dataset, under $\ell_\infty$ attack.

| Metric | PSNR | SSIM |
|---|---|---|
| MoDL | $15.01 \pm 5.46$ | $0.415 \pm 0.261$ |
| MoDL + DP (w/o FT) | $19.55 \pm 3.40$ | $0.585 \pm 0.144$ |
| MoDL + DP (w/ FT) | $20.14 \pm 3.34$ | $0.590 \pm 0.143$ |
| Proposed Method | $\mathbf{30.88 \pm 3.92}$ | $\mathbf{0.820 \pm 0.111}$ |

MoDL for the second stage. In other words, the following loss is used to fine-tune the MoDL, initialized from $\boldsymbol{\theta}$:

$$\arg\min_{\boldsymbol{\theta}_{\text{FT}}} \mathbb{E}\left[\mathcal{L}\big(f(\mathbf{z}_\Omega^{\text{pur}}, \mathbf{E}_\Omega; \boldsymbol{\theta}_{\text{FT}}), \mathbf{x}_{\text{ref}}\big)\right], \tag{30}$$

Finally, we generated $\ell_\infty$ perturbation in image domain on $\mathbf{z}_\Omega$ with 10 iterations of PGD (Mkadry et al., 2018), using $\epsilon = 0.01$. Following (Alkhouri et al., 2024), the final pipeline for reconstructing the perturbed input (i.e. $\mathbf{z}_\Omega^{\text{pert}}$) is as follows:

$$\mathbf{z}_\Omega^{\text{pur}} = (\mathbf{E}_\Omega^{\mathbf{H}} \mathbf{E}_\Omega) \boldsymbol{DP}_\phi(\mathbf{z}_\Omega^{\text{pert}}, \mathbf{E}_\Omega, N_r) \tag{31}$$

$$\hat{\mathbf{x}} = f(\mathbf{z}_\Omega^{\text{pur}}, \mathbf{E}_\Omega; \boldsymbol{\theta}_{\text{FT}}) \tag{32}$$

where $\hat{\mathbf{x}}$ is the final reconstruction. Fig. 17 shows the representative reconstructions. Tab. 15 summarizes the population metric on the test set. Note that the population metrics differ from those reported in the main text for Cor-PD, as this experiment uses a different undersampling pattern and a dataset with a matrix size of $320 \times 320$. Furthermore, when comparing MoDL+DP to MoDL itself, we observe an improvement of approximately 34% in PSNR, and 42% in SSIM, which is consistent with the findings reported in (Alkhouri et al., 2023) (where a 34% in PSNR and 19% in SSIM gain

was reported for the knee dataset in their Table 1). Comparing our baseline population metrics with those reported in their table (Alkhouri et al., 2023), we observe that our attack setup (strength and type) is stronger, leading to greater degradation of the baseline MoDL. Thus, the diffusion purification method results are quantitatively consistent with those reported in (Alkhouri et al., 2023), further highlighting the superior performance of our proposed approach.

## K ERROR EVALUATION ON $\{\Omega, \Omega^C\}$ SETS

As discussed in detail in Section 3.1, we expect the reconstruction to remain consistent with measurements on $\Omega$ locations, both in presence of an attack (as attack is designed to be small imperceptible perturbations on these lines) and in the absence of attack (a natural property of the PD-DL methods). Thus, the adversarial attack primarily corrupts the non-acquired lines (e.g. $\Omega^C$). We first demonstrated this intuition experimentally (Fig. 1a and Section 3.1), and also provided rigorous characterization with Theorem 1. We further corroborate these findings with an additional set of results depicted in Fig. 18.

In particular, for each knee sample from the test set, we generate the corresponding worst-case $\ell_\infty$ perturbation, and compute the final reconstructions, $\mathbf{x}$ and $\tilde{\mathbf{x}}$, of both the clean and perturbed inputs, respectively. We then map these onto the k-space locations specified by $\Omega$ and $\Omega^C$. Finally, we evaluate the $\ell_2$ error (normalized to clean data) on each of these index sets:

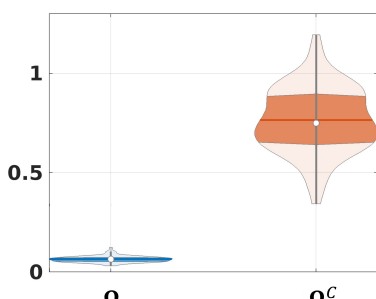

$$\frac{\|\mathbf{E}_\Omega(\mathbf{x} - \tilde{\mathbf{x}})\|_2}{\|\mathbf{E}_\Omega\mathbf{x}\|_2}, \qquad \frac{\|\mathbf{E}_{\Omega^C}(\mathbf{x} - \tilde{\mathbf{x}})\|_2}{\|\mathbf{E}_\Omega\mathbf{x}\|_2}, \qquad (33)$$

As Fig. 18 shows the reconstruction error on the $\Omega^C$ lines becomes extremely large with a small adversarial perturbation, while the error on $\Omega$ lines is comparatively negligible, owing to the data fidelity operations in the PD-DL network. As aforementioned, Theorem 1 serves the purpose of formally establishing these intuitions in a rigorous manner. Thus, it is complementary to the experimental insights that we provide here and in Section 3.1.

Figure 18: Normalized reconstruction errors on the $\Omega$ and $\Omega^C$ sets. The error on $\Omega^C$ is substantially larger, providing quantitative support for Theorem 1.

## L ADDITIONAL INVERSE PROBLEMS

It is worth investigating whether the proposed technique is useful in other modalities or inverse problems where PD-DL methods are used. First, we note that for non-physics-driven DL methods, since there is no data-

Table 16: Population metrics on the inpainting problem under the $\ell_\infty$ attack.

| Dataset | Metric | MoDL | Proposed + MoDL |
|---------|--------|------|-----------------|
| CelebA | PSNR | $24.05 \pm 1.07$ | $\mathbf{30.69 \pm 2.25}$ |
| | SSIM | $0.572 \pm 0.044$ | $\mathbf{0.873 \pm 0.034}$ |

consistency term, the expression $\mathbf{E}_\Omega\hat{\mathbf{x}}$ at different stages is not guaranteed to remain close to $\mathbf{y}_\Omega$ (e.g. measurements). As a result, cyclic consistency does not naturally extend to this setup. However, in cases where a data-fidelity term is used in the PD-DL network, then cyclic mitigation strategy can be utilized. In particular,

we further examine how the mitigation strategy extends to other inverse problem tasks, such as image inpainting. To this end, we used the same VSQP formulation as MoDL in the main text for image inpainting. The only modification was using a 3-channel input/output ResNet (as the dataset is natural images). The forward operator was a masking operator $\mathbf{B}_\Omega$. We randomly sampled 30% of the image as measurements, and trained the network on 1000 images from CelebA (Karras et al., 2017) dataset for 100 epochs with normalized $\ell_2$ loss. For the synthesized masks at inference, we used the remaining 70% of the pixels to generate two additional random $\Delta_i$, ensuring that none of the masks share any common pixel locations. We used 10 iterations of PGD for attack generation, with $\epsilon = 0.01 \cdot \|\mathbf{E}_\Omega^H\mathbf{y}_\Omega\|_\infty$, and $\alpha = \epsilon/5$, with projection into the $\epsilon$-ball applied after each PGD iteration. The representative reconstructions and population metrics over 200 test samples are represented in Fig. 19 and Tab. 16, respectively. These show that the proposed method effectively mitigates the

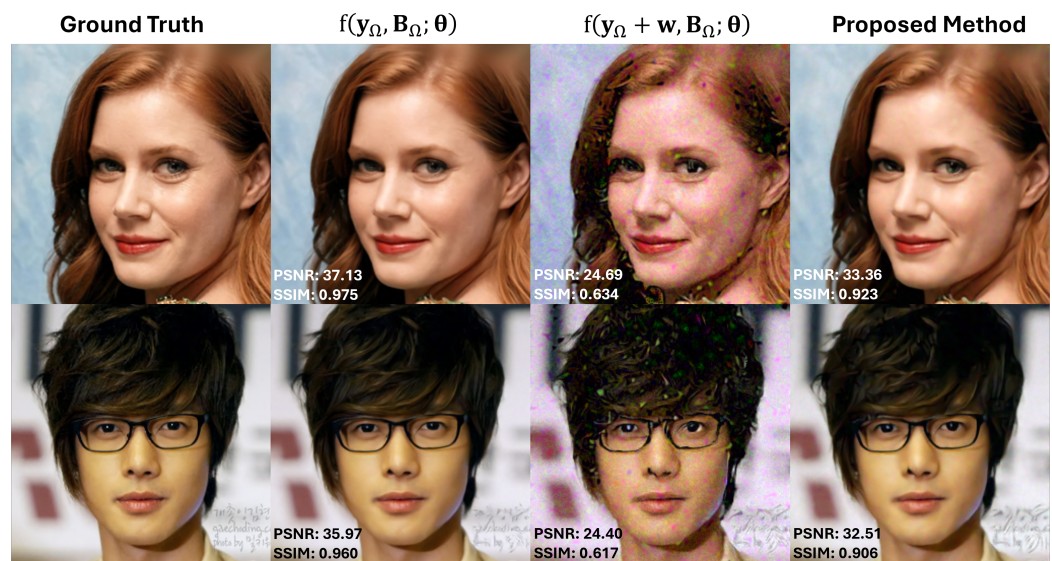

Figure 19: Representative results of random inpainting with 30% sampling.

perturbation and recovers the underlying image structure for the inpainting task on natural images, highlighting its versatility for PD-DL networks across applications.

## M   RUNTIME COMPARISON

In this section, we further analyze the runtime and computational costs of our method compared with other training-based defense approaches. Tab. 17 summarizes this comparison. The proposed method has a longer inference time because it requires multiple optimization iterations, each involving two forward passes through the network. However, when considering the combined training and inference time, our approach is substantially more efficient than the other two methods. This is because training-based defenses must retrain the network from scratch, and retune hyperparameters, if the attack configuration changes, whereas our method only requires tuning a few optimization parameters, which is far less expensive than full network retraining. Moreover, while we focused on 2D slices in the main text, our mitigation approach is not limited to 2D reconstructions. In particular, we evaluated the computational and memory costs of extending our method to dynamic MRI.

Table 17: Runtime comparison among AT, SMUG, and the proposed method.

| Method | Inference Time (sec) | Inference + Training Time (hours) |
|---|---|---|
| AT | 2.4 | $\sim 20$ |
| SMUG | 3.1 | $\sim 25.8$ |
| Proposed | 407 | $\sim 0.1$ (407 sec) |

Specifically, we performed dynamic cardiac MRI reconstruction using the OCMR dataset (Chen et al., 2020) and compared the per-iteration runtime and memory usage for 2D and dynamic settings. For the 2D case, we trained a standard MoDL model in a supervised manner using each cardiac phase as an independent sample, with input size $[1, N_x, N_y]$. For the dynamic case, we selected 10 consecutive time frames from each slice and concatenated them along the channel dimension, resulting in an input size of $[10, N_x, N_y]$. Both models were then used within our proposed mitigation framework under adversarial attack. The comparison in Table 18 shows that dynamic reconstruction does not im-

Table 18: Memory/time comparison of the proposed method.

| Strategy | Memory | Time per Iteration |
|---|---|---|
| 2D | 30.5 GB | 2.5 sec |
| Dynamic | 66.5 GB | 9.9 sec |

pose a substantial additional memory burden. While computation time naturally increases due to the higher input dimensionality, the overall scaling remains manageable and lower than the initial increase of the input size, indicating that our cyclic consistency–based defense extends naturally to higher-dimensional acquisitions.

# N    ROBUSTNESS UNDER NON-OPTIMAL RECONSTRUCTION CONDITIONS

In this section, we study how the proposed method performs under non-optimal conditions. Such conditions may arise in two ways: 1) When the baseline MoDL reconstruction is sub-optimal, and 2) When the perturbation is generated using a different forward model. For the former case, we note that the proposed method naturally inherits pre-trained model's limitations.

In other words, the mitigation strategy can, at best, recover the performance that the underlying model achieves on *clean inputs*, since we explore the vicinity of the adversary to find the non-perturbed version of it. This point was discussed in the manuscript when analyzing defense training methods, such as AT and SMUG (Section 4.2). For example, SMUG performs poorly under attack and fails to remove the resulting artifacts. After applying our mitigation approach, the artifacts are eliminated, but the reconstruction remains blurred (Fig. 2).

Table 19: Performance under $\ell_\infty$ attack with different pre-trained MoDL models. The suboptimal MoDL uses the weights from the $1^{st}$ epoch of training. Mean population metrics are reported.

| Dataset | Metric | Suboptimal MoDL | Proposed + Suboptimal MoDL | Proposed + MoDL |
|---------|--------|-----------------|----------------------------|-----------------|
| Cor-PD | PSNR | 18.04 | 30.19 | **35.14** |
|        | SSIM | 0.529 | 0.856 | **0.921** |

This limitation is rooted in the pre-trained SMUG model itself, because of smoothing with Gaussian noise. To provide further evidence supporting this concern, we applied the proposed method to very early-stage MoDL weights, which are not yet capable of fully recovering fine details. Tab. 19 reports the mean PSNR/SSIM of proposed mitigation algorithm with different pre-trained MoDLs. These results confirm that proposed mitigation approach improves the suboptimal reconstruction, without worsening the inherent sub-optimality of the baseline.

Table 20: Comparison of mitigation performance when the attack is generated using different coil sensitivity maps.

| Metric | Different Coil | Same Coil |
|--------|----------------|-----------|
| PSNR | 34.52 | 34.51 |
| SSIM | 0.911 | 0.907 |

The later case of non-optimality, however, is more complicated. One probable scenario, can be using a different sensitivity coil maps, to generate the perturbation. Under this mismatched coils (or forward operator), the generated attack will not be optimum. Since a mismatched coil configuration would not produce a good reconstruction for a given sample, the resulting attack would also no longer correspond to the true worst-case perturbation (though still within the $\epsilon$-ball), leading to a slightly milder adversarial effect. We tested this on a subset of Cor-PD dataset, with results shown in Tab. 20. In particular, we generated attacks using coil sensitivity maps from a different subject and then applied mitigation using the correct coil maps. Population metrics show that attacks generated with mismatched coil maps are slightly milder than those produced with the true coil configuration.

