# OpenReview forum: "Training-Free Defense Against Adversarial Attacks In Deep Learning MRI Reconstruction"
_ICLR.cc/2026/Conference — Submitted to ICLR 2026_

### Official Review · Reviewer_miBj · 2025-10-30

**Soundness:** 3
**Presentation:** 3
**Contribution:** 3
**Rating:** 6
**Confidence:** 5

**Summary:**

This paper introduces a training-free defense framework to mitigate adversarial perturbations in deep learning based MRI reconstruction, particularly for physics-driven deep learning (PD-DL) networks such as MoDL.

The proposed defense exploits cyclic measurement consistency (CMC): by resimulating undersampled k-space measurements from model reconstructions and enforcing self-consistency, the method detects and corrects adversarial perturbations without retraining or parameter modification.

The method formulates a reverse projected gradient descent (PGD) optimization in the input space to find a “corrective perturbation” that restores CMC. It is evaluated on fastMRI knee (Cor-PD) and brain (Ax-FLAIR) datasets, showing improvements over adversarial training and SMUG (smoothed unrolling). The method also generalizes to various PD-DL architectures and remains effective under blind and adaptive attacks.

**Strengths:**

The paper introduces a novel use of cyclic measurement consistency as a defensive objective, not just a training or calibration tool. This idea is well grounded in MRI physics and elegantly bridges signal reconstruction principles with adversarial robustness.

The approach works with any trained PD-DL model. This is a major advantage over retraining-based defenses like AT or SMUG. It can be combined with existing robust training for further gains.

Evaluated across multiple datasets, perturbation levels, and attack types. Also assessed on several architectures (MoDL, XPDNet, RIM, E2E-VarNet, Recurrent-VarNet), confirming generality.

Provides a theoretical analysis (Theorem 1) relating k-space perturbations and error propagation, lending physics-based interpretability.

**Weaknesses:**

The defense requires multiple forward passes through the reconstruction model (often dozens per iteration of reverse PGD). The runtime cost, though reported, may limit practical use in real-time MRI settings.

While strong within PD-DL MRI, the paper could discuss whether the approach generalizes to other imaging modalities or to non-physics-driven DL pipelines.

**Questions:**

na

---

> ### Author Response · Authors · 2025-11-22
>
> We sincerely appreciate the reviewer's encouraging and insightful feedback. We are glad that the novel use of cyclic measurement consistency as a defensive objective (not merely as a training or calibration tool) was recognized as a key contribution. We found it highly motivating that our approach was recognized to naturally build on MRI physics and bridge signal reconstruction principles with adversarial robustness. We also appreciate the recognition of the method’s compatibility with any trained PD-DL model, as well as its ability to complement existing robust-training strategies such as AT or SMUG. Finally, we value the positive comments on the theoretical analysis provided by Theorem 1, which offers a physically interpretable link between k-space perturbations and error propagation. We are grateful for the opportunity to clarify our contributions further and address the raised concerns.
>
> **(W1) Reverse PGD has a runtime cost that may not be suitable for real-time.**
>
> We thank the reviewer for drawing attention to this important aspect. We want to highlight that we do not claim real-time computation nor argue that real-time computation is necessary. As the reviewer points out, the proposed method requires multiple optimization iterations, each involving two forward passes through the network. Consequently, this results in higher computational cost compared to standard inference. Nonetheless, the method would still be useful in scenarios where the input leads to instability of the standard PD-DL reconstruction, as a means of making the acquired data usable, as rescanning the patient may not be always possible. In such scenarios, real-time computation may not necessarily be a driving factor.
>
> **(W2) Generalization to other imaging modalities or non-physics-driven DL pipelines.**
>
> We thank the reviewer for this excellent question regarding generalization. In non-physics-driven DL methods, since there is no data-consistency term, the expression $\mathbf{E_Ω} \mathbf{\hat{x}}$
> at different stages is not guaranteed to remain close to $\mathbf{y_Ω}$. As a result, Theorem 1 is no longer applicable, and cyclic consistency does not naturally extend to this setup. This highlights an advantage of PD-DL reconstruction, where data consistency is enforced within the network, allowing to leverage this property to design an effective cyclic-mitigation strategy against various types of perturbations.
>
> In principle, the idea can be extended to other modalities or inverse problems. The key assumption of Theorem 1 relies on characterizing the 2-norm of the matrix ${\bf AB}^H,$ where ${\bf A}$ is the original measurement matrix, and ${\bf B}$ is the measurement matrix for the synthesized measurementes at different locations. Thus, for most ``sampling'' based tasks, such as CT reconstruction, similar results can be derived. Unfortunately, we do not have expertise in this domain or access to datasets, thus we are unable to provide empirical results. Instead, we considered a similar inverse problem task for natural images, and implemented our approach for image inpainting. To this end, we used the same VSQP formulation as MoDL in the main text for image inpainting. The only modification was using a 3-channel input/output ResNet (as the dataset is natural images). The forward operator was a simple masking operator $\mathbf{B_Ω}$. We randomly sampled $\mathrm{30}$% of the image as measurements, and trained the network on 1000 images from CelebA (arXiv:1710.10196) dataset for 100 epochs with normalized $\ell_2$ loss. For the synthesized masks at inference, we used the remaining $\mathrm{70}$% of the pixels to generate two additional random $\Delta_i$, ensuring that none of the masks share any common pixel locations. We used 10 iterations of PGD for attack generation, with $\epsilon = 0.01.|| \mathbf{E_Ω}^\mathrm{H}\mathbf{y_Ω}||_\infty$, and $\alpha=\epsilon/5$. The population metrics are represented in the following table.
>
> **Table: Population metrics on the inpainting problem under the $\ell_\infty$ attack.**
>
> | Dataset | Metric | MoDL | Proposed Method + MoDL |
> |:-------:|:------:|:----:|:------------------------:|
> | **CelebA** | PSNR | 24.05 ± 1.07 | **30.69 ± 2.25** |
> |           | SSIM | 0.572 ± 0.044 | **0.873 ± 0.034** |

---

### Official Review · Reviewer_aJ4D · 2025-11-01

**Soundness:** 3
**Presentation:** 3
**Contribution:** 3
**Rating:** 6
**Confidence:** 5

**Summary:**

The paper proposes a training-free adversarial defense for physics-driven deep-learning (PD-DL) MRI reconstruction. Instead of retraining the model (as in Adversarial Training or SMUG), the authors introduce a cyclic measurement-consistency objective that perturbs the input slightly to restore reconstruction fidelity. A reverse-PGD procedure (Algorithm 1) iteratively corrects the attacked input by minimizing discrepancies between k-space projections of reconstructions from synthesized undersampling masks. The method is evaluated on fastMRI brain and knee datasets using MoDL and several other PD-DL architectures, and is shown to improve PSNR/SSIM under both standard and adaptive attacks.

**Strengths:**

- Thorough Evaluation: Multiple datasets (Cor-PD knee, Ax-FLAIR brain). Multiple architectures (MoDL, XPDNet, RIM, E2E-VarNet, Recurrent-VarNet). Multiple attack types: ℓ∞ image-domain, ℓ₂ k-space, sparse ℓ₀ (herringbone artifacts), and adaptive attacks.

- Physics-Driven Explanation: Theorem 1 offers an interpretable bound linking perturbations on acquired lines Ω to residuals on unacquired Ωᴄ, clarifying why cyclic inconsistency signals attacks.

- Strong Empirical Results: +1–3 dB PSNR gains over AT and SMUG; visual sharpness maintained.

**Weaknesses:**

1. While cyclic consistency is repurposed cleverly, much of the mathematical machinery (MoDL unrolling, PGD, consistency loss) builds directly on existing ideas; clarifying the conceptual leap beyond earlier “cycle-consistency” works (e.g., Zhang & Akçakaya 2024) would strengthen novelty.

2. Reverse-PGD plus multiple reconstructions is expensive (each iteration requires forward + inverse passes). Reported runtimes and wall-clock comparisons to AT/SMUG would help.

3. Paper should compare to concurrent defenses such as diffusion-based purification (Alkhouri et al., ICASSP 2024), as both are training-free physics-based methods.

**Questions:**

Please refer to the weakness section.

---

> ### Author Response · Authors · 2025-11-22
>
> We sincerely appreciate the reviewer's encouraging and detailed feedback. We are glad that the breadth of our evaluation was recognized as a strength. We also appreciate the acknowledgment of the physics-driven interpretation provided by Theorem 1, and its role in explaining why cyclic inconsistency effectively reveals perturbations on unacquired k-space lines. The reviewer's positive assessment of the empirical results, including the consistent PSNR gains over AT and SMUG while maintaining visual sharpness, is highly motivating. We are grateful for the opportunity to clarify our contributions further and address the raised concerns.
>
> **W1/Q1: The leap beyond earlier cycle-consistency works (e.g., Zhang \& Akçakaya 2024) should be clarified.**
>
> We thank the reviewer for highlighting this point. The cyclic-consistency approach has been used for training/calibrating reconstruction methods, for more than 15 years for parallel imaging methods (Zhao et al, 2008). For the more contemporary approaches, such as Zhang \& Akçakaya 2024, it is still used as primarily a training methodology for updating the reconstruction network. It uses a second-level of k-space at locations other than the acquired ones (using the same distribution), and incorporates them into an additional reconstruction step. These multiple reconstructions are then used to estimate mean and standard deviation of the reconstructions related to this PD-DL network. Subsequently, those are used to update the objective function the PD-DL solves, and a network to solve this objective function is retrained. Thus, conceptually while it uses the idea of data synthesis, it does not necessarily use ``consistency'' in the way we do in this paper. Furthermore, its theoretical results only concern the generalizability across different synthesized k-space locations, whereas ours are quite different, explicitly characterizing the effect of attacks on acquired and non-acquired k-space locations. We also respectfully disagree that our proof technique is limited to existing mathematical machinery used in MoDL unrolling. For instance, we explicitly bound the 2-norm of the matrix $\mathbf{E}_Ω\mathbf{E}_Ωᶜ ^H$, which requires physically-driven assumptions on coil maps (Lemma 1.1).
>
>    These theoretical results are then used in our proposed mitigation method to explicitly follow the PD-DL behavior on the acquired lines.  Even in the presence of an attack,  the final reconstruction remains consistent with the acquired data, which in turn forces any discrepancy to appear on the non-acquired lines. Here, we do use the same assumption as Zhang \& Akçakaya that a well-trained model generalizes to other sampling patterns drawn from the same distribution, but this idea is used in a completely novel way to formulate our objective accordingly.
>    In the presence of an attack, synthesizing measurements on the non-acquired lines Ωᶜ exposes the discrepancy introduced by the perturbation, enabling the proposed objective to explore the vicinity of the perturbed input and correct it.
>
> **W2/Q2: Computational cost of reverse-PGD?**
>
> We thank the reviewer for raising this question. Indeed reverse-PGD can be costly, but we note that other training-based approaches require retraining for each new attack scenario, and this retraining must be started with hyperparameters specifically tuned for the new attack setup. The table below reports the wall-clock inference and training times for different defense methods. As discussed, the proposed method has a higher inference cost due to reverse PGD, but compared with the substantial training burden required by other approaches, it can still be considered efficient overall. We have included this information in Appendix L.
>
> **Table: Runtime comparison among AT, SMUG, and the proposed method.**
>
> | Method   | Inference Time (sec) | Inference + Training Time (hours) |
> |:--------:|:---------------------:|:---------------------------------:|
> | **AT**       | 2.4   | ~20      |
> | **SMUG**     | 3.1   | ~25.8    |
> | **Proposed** | 407   | ~0.1 (407 sec) |

---

> ### Author Response · Authors · 2025-11-22
>
> **(W3) Comparison to concurrent training-free defenses like diffusion-based purification (Alkhouri et al., ICASSP 2024).**
>
> We thank the reviewer for this suggestion. For adversarial purification, we note that application of input purification directly to noisy zero-filled images is not straightforward. For instance, when considering diffusion purification, the setup here is not similar to the conventional classification-type tasks. In particular, diffusion models are trained to learn the distribution of clean fully-sampled data. Thus, if one noises the undersampled (aliased data) and tries to run the reverse diffusion process on this, it will inevitably push the output towards the clean data manifold, which will in turn no longer be consistent with the forward model (and the inverse problem objective). An early work (ICASSP-2024-10447906) considered diffusion purification for the MRI reconstruction problem studied here, and showed that even after purification, a fine-tuning step is required on MoDL network to handle the mismatch between the aliased images and score model trained on clean images. Subsequent work (arXiv:2309.05794) by the same authors further replaced the noising-denoising purification with a full diffusion model-based ScoreMRI reconstruction, combined with yet another fine-tuned MoDL network. Note furthermore that these works generate the adversarial attacks on a baseline MoDL model, and do not consider adaptive attacks that have knowledge about the ScoreMRI reconstruction. We have unfortunately not been able to successfully implement this method for comparison prior to our original submission. We also reached out to the authors of these papers, who graciously shared their code with us during the rebuttal period, but we have not been able to successfully reconstruct images with these pretrained models.
>     Thus, we instead cite and discuss this strategy in Appendix I without depicting experimental results that may be more sub-optimal to what the authors of these papers intended. In the same appendix, we additionally provide comparisons to other training-free input transforms, including randomized smoothing, JPEG compression and TV denoising, which we hope addresses some of the concerns of the reviewer.

---

### Official Review · Reviewer_M6zP · 2025-11-04

**Soundness:** 3
**Presentation:** 3
**Contribution:** 2
**Rating:** 4
**Confidence:** 3

**Summary:**

- The paper considers the task of mitigating adversarial attacks in PD-DL MRI reconstruction. The attack methods are assumed to be gradient-based approaches that find small perturbations within an $l_p$​ ball, applied to k-space or image observations, which lead to large deviations in the reconstruction output according to some metric (e.g., MSE).
- The paper proposes a method based on the cyclic consistency property of well-trained PD-DL networks to mitigate adversarial perturbations applied to the input image. The key idea is that adversarial perturbations break the cyclic consistency assumption of the trained PD-DL network. Therefore, minimizing the inconsistency of simulated reconstructed images can help recover the unperturbed image.

**Strengths:**

- The paper is well written, with a clear motivation for the proposed framework.
- The method is training-free, though it requires per-sample optimization.
- Experiments demonstrate the effectiveness of the proposed method against several standard adversarial attacks in MRI reconstruction.

**Weaknesses:**

- The need to mitigate adversarial attack in the domain of MRI reconstruction does not convince me.
- The proposed method relies on the assumption that the perturbation causes large changes in $\Omega^C$ and small changes in $\Omega$. However, quantitative justification for the bounds (lines 254–255) is missing. It is difficult to measure the effectiveness or generality of the proposed method across different scenarios.

**Questions:**

See weakness

---

> ### Author Response · Authors · 2025-11-22
>
> We sincerely appreciate the reviewer's positive and constructive feedback. We are glad that the clarity of the writing and the motivation for the proposed framework were well-received. The recognition of the benefits of a fully training-free formulation underscores a key strength of our contribution. We also appreciate the reviewer highlighting that the experiments effectively demonstrate robustness against multiple adversarial attacks in MRI reconstruction. We are grateful for the opportunity to clarify our contributions further and address the raised concerns.
>
>
>
> **(W1) Need for mitigating adversarial attacks in MRI reconstruction.**
>
> We thank the reviewer for this comment and agree that the motivation must be clearly articulated. In general, adversarial attacks are important for robustness evaluation, specifically in PD-DL reconstruction methods (as outlined in Section 2.4), as studying robustness under worst-case perturbations therefore provides a principled way to understand and mitigate milder, more realistic disturbances. In general, the perturbations of interest in the PD-DL setup are small input changes that break a PD-DL reconstruction without adversely affecting traditional linear reconstruction methods, since if they also break the traditional linear reconstruction method, it becomes an artifact issue that is independent of reconstruction technique.
>     Additionally, as discussed in Section 2.4, from a theoretical perspective, adversarial perturbations have a non-zero probability of occurrence (pnas.1907377117).  Finally, in the text, we further highlight a more realistic perturbation due to electromagnetic spikes that result in impulse-type noise in k-space, which lead to the well-known herringbone artifacts. While herringbone artifacts are normally visible when using linear reconstruction methods for high-intensity spikes, low-intensity spikes may still adversely affect the nonlinear PD-DL reconstructions, as we show in the manuscript. Our method is able to mitigate these realistic small-amplitude perturbations as well.

---

> ### Author Response · Authors · 2025-11-22
>
> **(W2) Quantitative justification for pertrubation being large in $\Omega^C$ and justification for lines 254-255.**
>
>
> First, for lines 254-255, we note that $||\mathbf{\tilde{x}}-\mathbf{x}||_2^2 = ||\mathbf{E_full} (\mathbf{\tilde{x}}-\mathbf{x})||_2^2$ by properties of the Fourier transform and the fact that root-sum-squares of coil maps is equal to identity. This leads to the decomposition in the text, $||\mathbf{E_full}(\mathbf{\tilde{x}}-\mathbf{x})||_2^2 = ||\mathbf{E}_Ω(\mathbf{\tilde{x}}-\mathbf{x}) + \mathbf{E}_Ωᶜ (\mathbf{\tilde{x}}-\mathbf{x})||_2^2 \xrightarrow{Ω\capΩᶜ = \varnothing} = || \mathbf{E}_Ω (\tilde{\mathbf{x}} - \mathbf{x}) \|_2^2 + || \mathbf{E}_Ωᶜ (\tilde{\mathbf{x}} - \mathbf{x}) ||_2^2$. Since $|| \mathbf{\tilde{x}} - \mathbf{x}||_2^2$ is large (due to the successful attack), at least one of the two terms on the right-hand side must be responsible for this increase. This ties in with how PD-DL methods alternate between proximal operations (implemented by neural networks) and data fidelity (DF) updates. Since the proximal operators primarily implement denoising, DF update is commonly placed at the end of each unrolled block. Thus, the full unrolled network concludes with a DF step that enforces data consistency. Accordingly, the reconstruction remains consistent with measurements on Ω locations, both in the presence of an attack (as attack is designed to be small imperceptible perturbations on these lines) and in the absence of attack (a natural property of the PD-DL methods). Thus, the adversarial attack primarily corrupts the non-acquired lines (e.g. Ωᶜ). Note in the text, we first demonstrated this intuition experimentally (Fig.1a and Section 3.1 in main text). We further corroborate this in Fig. 17 in Appendix J.) In particular, for each knee sample from the test set, we generated the corresponding worst-case ℓ∞ perturbation and computed the final reconstructions of both the clean and perturbed inputs, mapped on Ω and Ωᶜ locations. We then evaluated the $\ell_2$ error (normalized with clean data measurements) on each set of lines. As the figure shows, even under a small perturbation, the reconstruction error on the Ωᶜ lines becomes extremely large,  while the error on Ω lines is comparatively negligible. Theorem 1 serves the purpose of formally establishing these intuitions in a rigorous manner. Thus, it is complementary to the experimental insights that we provide. We have included this formulations in detail in Appendix J.

---

### Official Review · Reviewer_bQuX · 2025-11-06

**Soundness:** 2
**Presentation:** 2
**Contribution:** 2
**Rating:** 4
**Confidence:** 3

**Summary:**

This paper proposes a novel training-free defense method against adversarial attacks in physics-driven deep learning (PD-DL) MRI reconstruction. The approach leverages cyclic measurement consistency with synthesized under-sampling patterns to mitigate adversarial perturbations without retraining the model. The key idea is that adversarial attacks disrupt the consistency between reconstructions from actual and synthesized measurements, and this discrepancy can be minimized by optimizing a corrective perturbation within a small ball around the adversarial input. The authors demonstrate the effectiveness of their method across multiple datasets, attack types, PD-DL architectures, and realistic scenarios such as herringbone artifacts and adaptive attacks. The method also works in a blind setting without prior knowledge of attack strength or type.

**Strengths:**

(1) The method is well-motivated by the physics of MRI acquisition, particularly the role of data fidelity in unrolled networks and the propagation of perturbations to unacquired k-space locations.

(2) While cyclic consistency has been used in training and self-supervised learning, its adaptation for adversarial defense, especially in a training-free setting—is innovative.

**Weaknesses:**

(1) The adaptive attack results (Table 2) show that the iterative version of the defense remains effective, but the computational cost is glossed over. For T=100 unrolls, the iterative defense requires around 100 iterations—this is computationally prohibitive for real-time MRI reconstruction. The authors fail to analyze the trade-off between defense strength and computational efficiency.

(2) The paper mainly focuses on worst-case adversarial perturbations but does not test the method against realistic, non-malicious perturbations such as motion artifacts. A robust defense should also improve resilience to these common disturbances.

(3) The comparison is limited to defense methods that require retraining (adversarial training, SMUG). This fails to situate the method within the broader literature on training-free adversarial defenses, such as: (a) Input purification via denoising (e.g., diffusion models, Autoencoders); (b) Randomized smoothing or input transformations.

**Questions:**

(1) Theorem 1 does not clearly explain why the proposed mitigation works. It bounds the output error but does not directly justify the cyclic consistency objective. A more insightful analysis should connect the optimization landscape of cyclic loss to the removal of adversarial perturbations.

(2) The method heavily depends on generating realistic k-space data from previous reconstructions. If the PD-DL network has inherent biases or artifacts in its reconstructions, won't these be amplified through the cyclic process?  How does the performance degrade when the base reconstruction network itself is suboptimal or when there's a domain shift between training and test data? The defense seems to assume a nearly perfect base model.

(3) The current comparative analysis overlooks several training-free defense methods, such as diffusion-based purification, JPEG compression, and randomized smoothing. To provide a more comprehensive and fair evaluation, it is necessary to include at least two such baselines  with a clear discussion of the computational and performance trade-offs.

(4) The paper claims that this approach is physically driven, but the entire defense pipeline is conducted through the forward propagation of the deep learning network. Is this still essentially a "black box against black box" approach? If an attacker uses a completely different physical model (such as a different coil configuration) to generate an attack, is your defense mechanism still effective?

(5) All experiments appear to be on 2D MRI slices. How would the method be scaled to 3D volumes or dynamic MRI acquisitions where computational costs are substantially higher?  Does the cyclic consistency concept extend naturally to higher dimensions?

---

> ### Author Response · Authors · 2025-11-21
>
> We sincerely appreciate the reviewer's thoughtful and encouraging feedback. We are glad that our motivation was found to be grounded in MRI physics; and in particular the role of data fidelity in unrolled networks and the propagation of perturbations into unacquired k-space were recognized as strengths. We also appreciate the positive assessment of the novelty of adapting cyclic consistency for adversarial defense. We are grateful for the opportunity to clarify our contributions further and address the raised concerns.
>
> **(W1) Iterative defense is computationally expensive, not suitable for real-time.**
>
> We thank the reviewer for bringing attention to this important aspect. Indeed, defense against adaptive attacks is the hardest scenario to handle requiring many iterative steps. But we want to highlight that we do not claim real-time computation nor argue that real-time computation is necessary. As the reviewer points out, the proposed method requires multiple optimization iterations, each involving two forward passes through the network. Consequently, this results in higher computational cost compared to standard inference. Nonetheless, the method would still be useful in scenarios where the input leads to instability of the standard PD-DL reconstruction, as a means of making the acquired data usable, as rescanning the patient may not be always possible. In such scenarios, real-time computation may not necessarily be a driving factor.
>
>
> **(W2) Method tested on worst-case adversarial attacks, and not on non-malicious perturbations like motion.**
>
> As discussed in Section 2.4, worst-case perturbations are important to understand the robustness of PD-DL reconstruction methods in general. While motion artifacts arise naturally, these are harder to model accurately as mentioned in the main text. Thus, studying the robustness under worst-case attack constraints give us a principled way to understand and mitigate more realistic but potentially milder perturbations, such as those related to motion. Since motion artifacts are difficult to model precisely, we do not consider these, but instead study small worst-case adversarial perturbations within a small $\ell_p$ ball around the input. Note that in general, the perturbations of interest in the PD-DL setup are small input changes that break a PD-DL reconstruction without adversely affecting traditional linear reconstruction methods, since if they also break the traditional linear reconstruction method, it becomes an artifact issue that is independent of reconstruction technique. Thus, in this setup, a small motion-related artifact, which would break a PD-DL reconstruction without adversely affecting traditional linear reconstruction methods, could be modeled as a perturbation within a similar $\ell_p$ ball, even if it is a milder perturbation than the worst-case one considered here.
>
>
> In the text, we further highlight a more realistic perturbation due to electromagnetic spikes that result in impulse-type noise in k-space, which lead to the well-known herringbone artifacts. While herringbone artifacts are normally visible when using linear reconstruction methods for high-intensity spikes, low-intensity spikes may still adversely affect the nonlinear PD-DL reconstructions, as we show in the manuscript. Our method is able to mitigate these realistic small-amplitude perturbations as well.

---

> ### Author Response · Authors · 2025-11-22
>
> **W3 \& Q3: Comparisons to other training-free defenses, (a) input purification, (b) randomized smoothing/input transformations, would be helpful.**
>
> We thank the reviewer for this insightful feedback. We will answer the two parts separately.
>
> For (a) input purification, we note that application of input purification directly to noisy zero-filled images is not straightforward. For instance, when considering diffusion purification, the setup here is not similar to the conventional classification-type tasks. In particular, diffusion models are trained to learn the distribution of clean fully-sampled data. Thus, if one noises the undersampled (aliased data) and tries to run the reverse diffusion process on this, it will inevitably push the output towards the clean data manifold, which will in turn no longer be consistent with the forward model (and the inverse problem objective). An early work (ICASSP-2024-10447906) considered diffusion purification for the MRI reconstruction problem studied here, and showed that even after purification, a fine-tuning step is required on MoDL network to handle the mismatch between the aliased images and score model trained on clean images. Subsequent work (arXiv:2309.05794) by the same authors further replaced the noising-denoising purification with a full diffusion model-based ScoreMRI reconstruction, combined with yet another fine-tuned MoDL network. Note furthermore that these works generate the adversarial attacks on a baseline MoDL model, and do not consider adaptive attacks that have knowledge about the ScoreMRI reconstruction.
>
> For (b) we implemented randomized smoothing (RS), as well as input transformations using JPEG compression and total variation (TV) denoising, on the input as training-free defenses. For RS, we generated 50 zero-mean Gaussian samples $\mathbf{\xi}\sim\mathcal{N}(0,\lambda\mathbf{I})$, and added them to the input. Each perturbed sample was then reconstructed using baseline MoDL, and the final output was obtained by averaging the reconstructions. Further implementation details about tuning λ are presented in the Appendix I. For JPEG compression, we first split the complex input into four channels corresponding to the positive and negative components of the real and imaginary parts. Each channel was normalized to the [0,255] range, and JPEG compression was applied. Finally, for TV denoising, we solved the denoising problem $min_\mathbf{\mathrm{z}}||\mathbf{\mathrm{z}}_\mathrm{\Omega}−\mathbf{\mathrm{z}}||_2^2+β·TV(\mathbf{\mathrm{z}})$,
> where $TV(.)$ denotes the total-variation regularization term. We used $\mathrm{\beta}=0.01$, and performed 5 optimization iterations, since additional iterations tend to de-alias $\mathbf{𝐳}_Ω$, similar to what was discussed above for diffusion purification. Further implementation details about attack generation is presented in Appendix I. Population metrics across the test set are presented in the following table (also Tab. 13 in the Appendix):
>
> **Table: Comparison of different input-transformation defenses on the Cor-PD dataset under perturbation.
> Population mean metrics are reported.**
>
> | Dataset | Metric | MoDL | Input TV + MoDL | Input JPEG + MoDL | Input RS + MoDL | Proposed + MoDL |
> |:-------:|:------:|:----:|:----------------:|:------------------:|:----------------:|:----------------:|
> | **Cor-PD** | PSNR | 16.30 | 16.36 | 17.29 | 30.71 | **35.14** |
> |           | SSIM | 0.486 | 0.488 | 0.509 | 0.858 | **0.921** |
>
> Both methods fail to mitigate the attack in a meaningful manner. RS outperforms these two input transformation methods, but still substantially falls short of the proposed method. Representative mitigated reconstructions using these approaches are shown in Fig. 16 in Appendix.

---

> ### Author Response · Authors · 2025-11-22
>
> **(Q1) Theorem 1 bounds output error but does not directly justify cyclic consistency.**
>
> We thank the reviewer for pointing this out for further clarity. PD-DL methods for inverse problems (e.g. MRI reconstruction, inpainting, phase retrieval) alternate between proximal operations (implemented by neural networks) and data fidelity (DF) updates. Since the proximal operators primarily implement denoising, DF update is commonly placed at the end of each unrolled block. Thus, unrolled network concludes with a DF step that enforces data consistency. Accordingly, we expect the reconstruction to remain consistent with measurements on $\Omega$ locations, both in presence of an attack (as attack is designed to be small imperceptible perturbations on these lines) and in absence of attack (a natural property of the  PD-DL methods). Thus, the adversarial attack primarily corrupts the non-acquired lines (e.g. $\Omega^C$)
>
> Note in the text, we first demonstrated this intuition experimentally (Fig.1a and Section 3.1 in main text). We further corroborate this with an additional Fig. 17 in Appendix J. In particular, for each knee sample from the test set, we generated the corresponding worst-case $\ell_\infty$ perturbation and computed the final reconstructions of both the clean and perturbed inputs, mapped on $\Omega$ and $\Omega^C$ locations. We then evaluated the $\ell_2$ error (normalized with clean data measurements) on each set of lines. As the figure shows, even under a small perturbation, the reconstruction error on the $\Omega^{C}$ lines becomes extremely large,  while the error on $\Omega$ lines is comparatively negligible. Theorem 1 serves the purpose of formally establishing these intuitions in a rigorous manner. Thus, it is complementary to the experimental insights that we provide.
>
> These insights then motivate the cyclic-consistency idea, as detailed in Section 3.2, where the mitigation algorithm exploits this property (reconstruction after adversarial attack remains consistent on $\Omega$ locations, but is corrupted on non-acquired $\Omega^C$ lines). We devised the cyclic-consistency objective such that the first stage reconstructs the image using the $\Omega$ measurements, and the second stage synthesizes measurements from this reconstruction on a subset of non-acquired lines, $\Delta_i \subset \Omega^C$. Because the optimization is performed over a corrective perturbation $\mathbf{r}'$, only perturbations that remain consistent across this cycle can minimize the objective. Consequently, the resulting solution cannot introduce large discrepancies on either the $\Omega$ or $\Omega^C$ lines, and since the optimization explores the local neighborhood of the perturbed image, the resulting solution is close to the clean input.
>
> **(Q2) Dependence on the quality of the pre-trained PD-DL network.**
>
> Our proposed method naturally inherits pre-trained model's limitations. In other words, the mitigation strategy can, at best, recover the performance that the underlying model achieves on *clean inputs*, since we explore the vicinity of the adversary to find the non-perturbed version of it. This point was discussed in the manuscript when analyzing defense training methods, such as AT and SMUG. For example, SMUG performs poorly under attack and fails to remove the resulting artifacts. After applying our mitigation approach, the artifacts are eliminated, but the reconstruction remains blurred. This limitation is rooted in the pre-trained SMUG model itself, because of smoothing with Gaussian noise.
>
> To provide further evidence supporting this concern, we applied the proposed method to very early-stage MoDL weights, which are not yet capable of fully recovering fine details. The following table reports the mean PSNR/SSIM of proposed mitigation algorithm with different pre-trained MoDLs. These results confirm that proposed mitigation approach improves the suboptimal reconstruction, without worsening the inherent sub optimality.
>
> **Table: Performance under $\ell_{\infty}$ attack with different pre-trained MoDL models.
> The suboptimal MoDL uses the weights from the $1^\text{st}$ epoch of training.
> Population mean metrics are reported.**
>
> | Dataset | Metric | Suboptimal MoDL | Proposed + Suboptimal MoDL | Proposed + MoDL |
> |:-------:|:------:|:----------------:|:---------------------------:|:----------------:|
> | **Cor-PD** | PSNR | 18.04 | 30.19 | **35.14** |
> |           | SSIM | 0.529 | 0.856 | **0.921** |

---

> ### Author Response · Authors · 2025-11-22
>
> **(Q4) How is the approach physically-driven?**
>
> Thank you for this question. The physics information is inherently incorporated through the data fidelity units, as in standard PD-DL networks. This is also the main idea behind cyclic-consistency, as detailed in Q1. Specifically, our approach computes gradients by propagating through the entire network as well as the cyclic encoding operators twice. Therefore, it incorporates the MRI physics through the data-fidelity modules in the unrolled network, while also including the forward propagation within the cyclic consistency loss.
>
> For the second part of the question regrading coil maps, we note that under mismatched coils (or forward operator) the generated attack will not be optimum. Since a mismatched coil configuration would not produce a good reconstruction for a given sample, the resulting attack would also no longer correspond to the true worst-case perturbation (though still within the $\epsilon$-ball), leading to a slightly milder adversarial effect. We tested this on a subset of Cor-PD dataset, with results shown in the following table. In particular, we generated attacks using coil sensitivity maps from a different subject and then applied mitigation using the correct coil maps, as the reviewer suggested. Population metrics show that attacks generated with mismatched coil maps are slightly milder than those produced with the true coil configuration.
>
> **Table: Comparison of mitigation performance when the attack is generated with different coil sensitivity maps.**
>
> | Metric | Different Coil | Same Coil |
> |:------:|:--------------:|:---------:|
> | **PSNR** | 34.52 | 34.51 |
> | **SSIM** | 0.911 | 0.907 |
>
> **(Q5) How would the method scale to 3D or dynamic MRI?**
>
> We thank the reviewer for pointing this out. The method itself is directly applicable to 3D or dynamic MRI, though we understand the reviewer's concern about computational costs. To address this concern, we evaluated the computational and memory costs of extending our method to dynamic MRI. Specifically, we performed dynamic cardiac MRI reconstruction using the OCMR dataset (arXiv:2008.03410) and compared the per-iteration runtime and memory usage for 2D and dynamic settings. For the 2D case, we trained a standard MoDL model in a supervised manner using each cardiac phase as an independent sample, with input size $\mathrm{[1,N_x, N_y]}$. For the dynamic case, we selected 10 consecutive time frames from each slice and concatenated them along the channel dimension, resulting in an input size of $\mathrm{[10,N_x, N_y]}$. Both models were then used within our proposed mitigation framework under adversarial attack.
> The comparison in the following table shows that dynamic reconstruction does not impose a substantial additional memory burden. While computation time naturally increases due to the higher input dimensionality, the overall scaling remains manageable and lower than the initial increase of the input size, indicating that our cyclic consistency–based defense extends naturally to higher-dimensional acquisitions.
>
> **Table: Memory/time comparison of proposed method.**
>
> | Strategy | Memory | Time per Iteration |
> |:--------:|:------:|:------------------:|
> | **2D**      | 30.5 GB | 2.5 sec |
> | **Dynamic** | 66.5 GB | 9.9 sec |

---

### Author Response · Authors · 2025-12-03
**Final Remarks**

We sincerely thank all the Reviewers, Area Chairs, Senior Area Chairs, and Program Chairs for the time, expertise, and constructive feedback devoted to our submission. We are encouraged by the reviewers’ positive recognition of our work, highlighting the clear motivation and strong physics-driven foundations of our approach (**all reviewers**). The reviewers highlighted the generality of the method and its thorough evaluation (**M6zP, aJ4D, miBj**), emphasized the novelty of adapting cyclic consistency into a defensive objective (**bQuX, miBj**), appreciated the adaptability of our approach to any PD-DL method, as well as its fully training-free nature (**miBj, M6zP**), and valued the strong empirical results across multiple experimental settings (**M6zP, aJ4D, miBj**).

During the rebuttal period, we thoroughly addressed the reviewers’ concerns, which further explored additional strengths of our method:

**Further justification of the proposed method.** Our mitigation idea is grounded in MRI physics (Sec. 3.1), and is further supported by Theorem 1 and its accompanying lemmas (Appendix G). During the rebuttal, we highlighted the new mathematical machinery in our proofs, for instance related to the physics of multi-coil MRI acquisitions (Lemma 1.1). Moreover, we quantitatively validated the bounds established by Theorem 1, demonstrating that non-acquired lines are corrupted substantially more than acquired lines as theoretically predicted (Appendix K, Fig. 18).


**Comparison to other training-free methods.** We added comparisons to several training-free defenses, such as input transformations (JPEG compression and TV denoising) and randomized smoothing (Fig. 16, Tab. 14), showing our approach substantially outperforms these. We also added comparisons to diffusion-based purification *since our last response*, once again confirming the intuition that diffusion purification is not directly applicable to aliased inputs in MRI reconstruction, requiring fine-tuning of the networks (thus not training-free), while still being substantially outperformed by our method (Fig. 17, Tab. 15). Details are in Appendix I and J, respectively.

**Runtime analysis.**
We added runtime comparisons between our method and training-based defenses, highlighting the advantages of its training-free design (Appendix M, Tab. 17), when considering the combined computational cost of training and inference.

In addition, we examined how the computational cost scales in higher dimensions, showing that our method on dynamic MRI (2D + time) does not impose a substantial additional memory burden over simple 2D MRI. Though the computation time naturally increases due to larger data size, the overall scaling remains manageable and lower than the initial increase of the input size, indicating that our defense extends naturally to higher-dimensional datasets (Appendix M, Tab. 18).



**Importance of adversarial attacks in MRI reconstruction.** We expanded our discussion on this importance (Section 2.4) to address a comment from Reviewer **M6zP**. Other than the theoretical perspective readily discussed in the initial submission, we highlighted the importance of electromagnetic spikes, especially in ultrahigh field MRI, which result in impulse-type noise in k-space. While herringbone artifacts arising from such impulse noise are normally visible in linear reconstruction for high-intensity spikes, low-intensity spikes may still adversely affect the nonlinear PD-DL reconstructions as we show *for the first time* (Fig. 5, Appendix B). Our method remains effective in this scenario. At Reviewer **bQuX**'s request for a scenario including coil mismatches, we also performed experiments, showing that our method still successfully mitigates such attacks (Appendix N).


**Additional experiments.** These include evaluations of our method on a PD-DL solution for inpainting on natural images as an example of a different inverse problem. Note our method generalizes naturally to this setup, owing to the physics-driven nature of our mitigation approach (Appendix L, Fig. 19, Tab. 16). We also evaluated our method when the attack is non-optimal or when the pre-trained model is suboptimal. In both scenarios, our approach remained consistently effective (Appendix N, Tables 19-20).

Unfortunately, we could not engage with all reviewers due to the OpenReview incident. Nonetheless, we fully addressed their comments in our rebuttal and believe our clarifications resolve the concerns raised.

We appreciate the constructive engagement during rebuttal, and hope the final decision reflects the positive developments and strengthened evidence for our novel physics-guided mitigation approach for adversarial attacks in MRI reconstruction with practical applications.

Warm regards,

Authors of submission 19159

---

### Meta-Review · Area_Chair_pqTj · 2026-01-06

**Summary:**

This paper presents a method for defense against potential attack to MRI reconstruction. As pointed out by the reviewers, the motivation of the attack is not well motivated and the authors shall focus on realistic attacks. In fact, attacking to medical imaging data is not realistic unless such attach happens in the data acquisition. Therefore, the authors shall come from this angle. Unfortunately, this is not the case for this paper. Although the authors provide some experiments on some possible attacks, I believe a significant revision or even re-written of the paper is needed. Moreover, the evaluation of such reconstruction shall be conducted on both data with and without lesions. Therefore, the evaluation shall consider how the approaches help in this perspective.

**Reviewer Concerns:**

The reviewers concerns are partially addressed but I believe this paper needs a major rewrite.

**Reviewer Scores:**

The reviewers are likely to maintain their scores.

---

### Decision · Program_Chairs · 2026-01-26

Reject